# On-surface lithium donor reaction enables decarbonated lithium garnets and compatible interfaces within cathodes

Ya-Nan Yang[1,2], Ying-Xiang Li[1,2], Yi-Qiu Li[1] & Tao Zhang [1,2✉]

Lithium garnets have been widely studied as promising electrolytes that could enable the next-generation all-solid-state lithium batteries. However, upon exposure to atmospheric moisture and carbon dioxide, insulating lithium carbonate forms on the surface and deteriorates the interfaces within electrodes. Here, we report a scalable solid sintering method, defined by lithium donor reaction that allows for complete decarbonation of $Li_{6.4}La_3Zr_{1.4}Ta_{0.6}O_{12}$ (LLZTO) and yields an active $LiCoO_2$ layer for each garnet particle. The obtained $LiCoO_2$ coated garnets composite is stable against air without any $Li_2CO_3$. Once working in a solid-state lithium battery, the $LiCoO_2$-LLZTO@$LiCoO_2$ composite cathode maintains 81% of the initial capacity after 180 cycles at 0.1 C. Eliminating $CO_2$ evolution above 4.0 V is confirmed experimentally after transforming $Li_2CO_3$ into $LiCoO_2$. These results indicate that $Li_2CO_3$ is no longer an obstacle, but a trigger of the intimate solid-solid interface. This strategy has been extended to develop a series of LLZTO@active layer materials.

[1] State Key Lab of High Performance Ceramics and Superfine Microstructure, Shanghai Institute of Ceramics, Chinese Academy of Sciences, 1295 Dingxi Road, Shanghai 200050, P.R. China. [2] Center of Materials Science and Optoelectronics Engineering, University of Chinese Academy of Sciences, Beijing 100049, P.R. China. ✉email: taozhang@mail.sic.ac.cn

Solid-state batteries (SSBs) with a high-capacity lithium metal anode are considered as the ultimate alternative to liquid lithium-ion batteries[1], which not only exhibit higher energy density but also fundamentally solve the safety problems of the liquid batteries due to the utilization of non-flammable solid-state electrolytes (SSEs). However, the solid–solid interfaces between SSEs and electrodes cause a large inherent impedance in the SSBs. At the same time, the SSEs are completely non-wetting compared with the liquid electrolyte so that the electrolyte cannot be immersed in the cathode to construct lithium-ion transport pathways, slowing the diffusion of lithium ions between particles inside the cathode. In order to address this issue, solid electrolyte powders are often added to the cathodes and the interface between solid electrolytes and active materials is designed to increase the ionic conductivity and decrease the polarization of electrode[2–5]. In the recent year, the SSBs employing a garnet-structured $Li_7La_3Zr_2O_{12}$ (LLZO) electrolyte have shown significant promise in practical applications because the LLZO electrolyte has high lithium-ion conductivity and is stable to lithium metal, but again, the ionic conductivity inside the cathode is low due to the use of the non-wetting LLZO solid electrolyte piece. Some efforts have been made to build lithium-ion transport channels inside cathode to preparing high performance composite cathodes in LLZO-based SSBs. For instance, Wakayama et al. reported a three-dimensional bicontinuous composite cathode which increased the surface area of the interface between the active materials and the LLZO particles[6]. Broek et al. embedded the electrode materials to the porous LLZO electrolyte, and it is beneficial to converting of lithium ions inside the electrode[7]. Besides, the interface properties of the composite cathode can also be improved by forming a coated structure in which active materials are coated with the LLZO particles[8].

Unfortunately, it has been reported that LLZO is unstable in moist air and it is spontaneous to react with $H_2O$ and $CO_2$ to generate a $Li_2CO_3$ layer on the surface[9]. The $Li_2CO_3$ layer is lithiophobicity and has an ultralow low lithium-ion conductivity so that it is one of the sources of the high interfacial impedance in SSEs[10,11]. So far, although it has been reported that $Li_2CO_3$ on the surface of LLZO can be removed by surface polishing[11] or chemical reaction[12], these approaches based on "eliminating" concept have just short-term effectiveness and in particular, only suitable for handling the large-sized surface of electrolyte piece. In contrast, a method for removing the $Li_2CO_3$ layer on the surface of the LLZO powder that has a larger surface area with more $Li_2CO_3$ has not been reported. This hinders the fast transport of lithium-ions inside cathode when adding LLZO powder to the cathode as an ion conductor or designing the internal interface inside the cathode. Therefore, reliable solutions to remove the $Li_2CO_3$ layer and to establish intimate physical contact between LLZO and active cathode materials are still needed.

Herein, we propose an "interface homogeneity" strategy to transform the $Li_2CO_3$ into $LiCoO_2$ active material on the surface of $Li_{6.4}La_3Zr_{1.4}Ta_{0.6}O_{12}$ (LLZTO) by an on-surface lithium-donor reaction. Significantly, the LLZTO coated with $LiCoO_2$ (LLZTO@LCO) was obtained by the reaction of the $Li_2CO_3$ layer on the surface of LLZTO with $Co_3O_4$. The transformation from $Li_2CO_3$ into $Li_2CoO_2$ is complete and not reversible, indicating that the $Li_2CO_3$ layer can be fully removed. The formed $LiCoO_2$ layer ensures direct contact between the solid electrolyte particles and the homogeneous $LiCoO_2$ cathode material, circumventing the conventional heterogeneous solid–solid interface problem inside composite cathodes. In this work, the proposed LLZTO@LCO materials were successfully synthesized and characterized. For comparison, the LLZTO coated with naturally formed $Li_2CO_3$ (LLZTO@$Li_2CO_3$) and the LLZTO@LCO were

used as an ionic conductor to prepare composite cathodes with $LiCoO_2$ active materials, separately. And then LLZTO-based SSBs were assembled. We found that the battery with an LLZTO@LCO-containing $LiCoO_2$ composite cathode exhibited a high Coulombic efficiency (CE) and improved cycling performance.

## Results

**Characterization of LLZTO and LLZTO@LCO.** The LLZTO@LCO materials were prepared by a two-step solid sintering process. Figure 1a shows the transmission electron microscopy (TEM) image of LLZTO after air exposure for 4 weeks. It can be seen that a 0.1 μm thick layer formed on the surface of LLZTO. The coating layer can be indexed to the monoclinic $Li_2CO_3$ by subjecting selected area electron diffraction of the encircled region (Fig. S1). Figure 1b shows the scanning electron microscopy (SEM) image of LLZTO, the particle size of LLZTO is about 4 μm and has a smooth surface as well as irregular shape. After the reaction, instead of $Li_2CO_3$ layer, $LiCoO_2$ is evenly distributed on the surface of LLZTO to form a coating with a thickness in the range of 0.3–0.5 μm, as shown in Fig. 1c. The inset in Fig. 1c shows the high resolution TEM (HRTEM) image of the encircled region. The lattice spacing of 0.245 nm agrees with the (101) facets of crystallized $LiCoO_2$, and the widely exposed (101) facets (Fig. S2) exhibit higher ionic conductivity and electrochemical activity[13]. Moreover, as Fig. 1d shows, the LLZTO@LCO exhibits a spherical structure, which can increase the stacking density of composite cathode. The $LiCoO_2$ exhibits a nanoplate-like, which is beneficial to the rate performance of the battery[14]. Figure 1e shows the energy-dispersive X-ray mapping analysis results. The Co, O elements are uniformly distributed on the surface of particles and the La, Zr elements can also be detected in that region, corresponding to the LLZTO@LCO structure. The X-ray diffraction (XRD) patterns, as shown in Fig. 1f, are in agreement with the TEM and SEM results. Compared with $Li_5La_3Nb_2O_{12}$ PDF card (45-0109), the existence of $Li_2CO_3$ on the surface of LLZTO was confirmed before reaction. Apparently, the ultimate materials only contain LLZTO and $LiCoO_2$ after the reaction, and the $LiCoO_2$ coating has R-3m symmetry as the traditional $LiCoO_2$ active materials[15]. Above results confirm that the transformation of $Li_2CO_3$ to $LiCoO_2$ is well achieved, but we find that an impurity is also formed during the first sintering. Figure 1g shows the XRD pattern of the lithium-donor reaction products after the first sintering. It should be noted that the impurity is $La_2Zr_2O_7$. This is due to the volatilization of lithium from LLZTO during sintering[16]. Besides, after first sintering, the substitution of $Li_2CO_3$ by $LiCoO_2$ indicates that the $Li_2CO_3$ coating has been fully reacted with $Co_3O_4$ to generate $LiCoO_2$, but the $LiCoO_2$ exhibits a block-like rather than a nanoplate-like (Fig. S3).

**Reaction mechanism of transforming LLZTO@$Li_2CO_3$ to LLZTO@LCO.** As shown in Fig. 2a, transforming LLZTO@-$Li_2CO_3$ to LLZTO@LCO was achieved by the on-surface lithium-donor reaction. By sintering different rations of $Co_3O_4$ and LLZTO@$Li_2CO_3$, we found that there is no $Co_3O_4$ left after the excessive $Co_3O_4$ reacts with a small amount of $Li_2CO_3$ from the surface of LLZTO (Fig. S4). This is because, in addition to the reaction of $Li_2CO_3$ with $Co_3O_4$, there is also a reaction between lithium volatiles from LLZTO and the remaining $Co_3O_4$ to generate $LiCoO_2$. $Li_2CO_3$ and LLZTO as the lithium donors together provide the lithium sources for the transformation reaction to form the complete $LiCoO_2$ coating on the surface of LLZTO. To confirm the above process, we designed two experiments. First, $Co_3O_4$ with $Li_2CO_3$ materials were sintered under the first

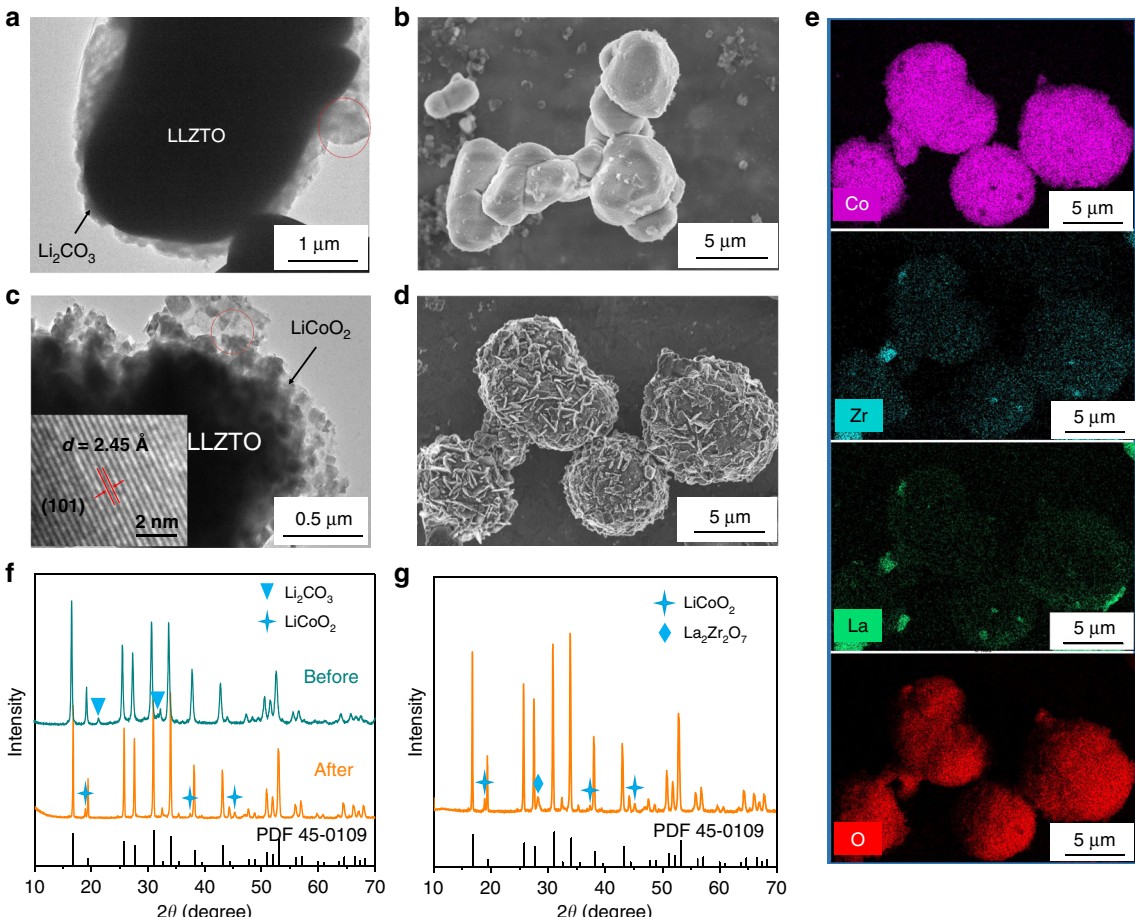

**Fig. 1 Characterization of LLZTO@Li$_2$CO$_3$ and LLZTO@LCO. a, b** TEM image and SEM image of LLZTO powder exposed to air for four weeks. **c, d** TEM image and SEM image of LLZTO@LCO. **e** EDX mapping analysis of LLZTO@LCO corresponding to (**d**). **f** XRD patterns of LLZTO@Li$_2$CO$_3$ and LLZTO@LCO. **g** XRD pattern of materials after first sintering containing impurity.

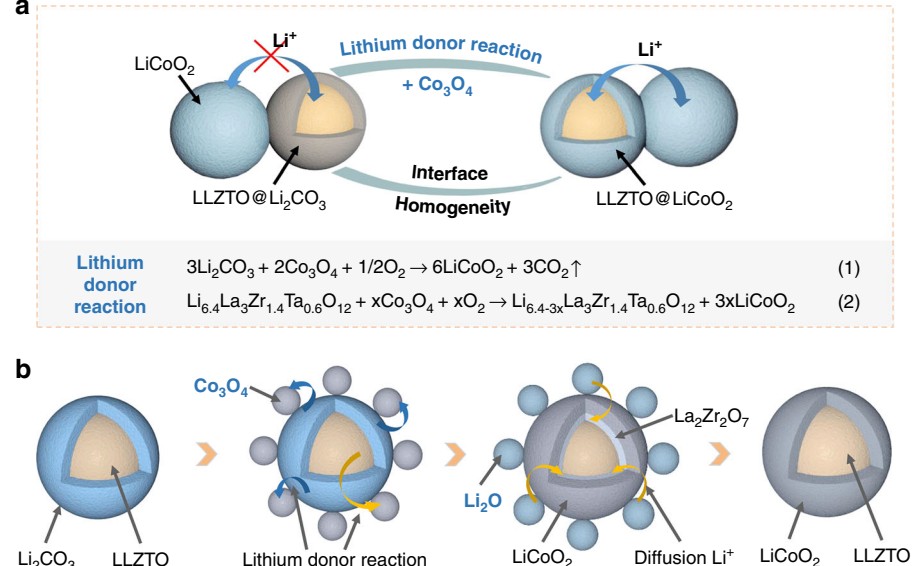

**Fig. 2 Process of transforming LLZTO@Li$_2$CO$_3$ into LLZTO@LCO. a** Schematic illustration of the lithium-donor reaction to achieve interface homogeneity. **b** Schematic illustration of the two-step solid state reaction process of transforming LLZTO@Li$_2$CO$_3$ into LLZTO@LCO.

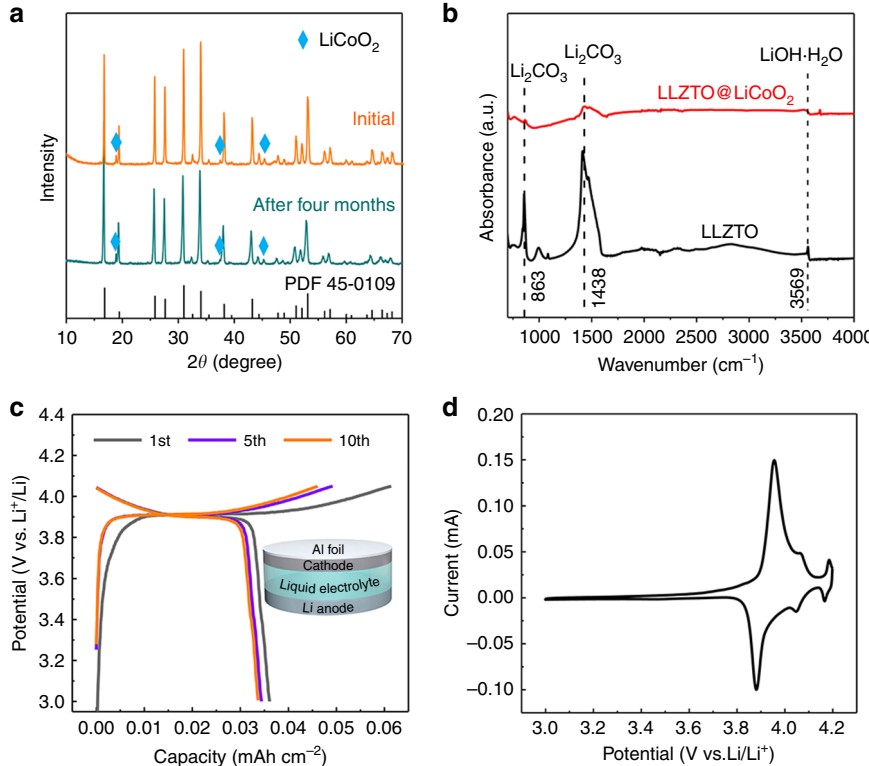

**Fig. 3 Stability and activity test of LLZTO@LCO. a** XRD patterns of LLZTO@LCO before and after exposure to air for four months. **b** FTIR spectra of the LLZTO and LLZTO@LCO samples after exposure to air for four months. **c, d** Charge/Discharge curves and cyclic voltammetry profile of the liquid cells. The inset in Fig. 3c shows the structural illustration of the cell, in which the cathode was prepared by mixing LLZTO@LCO, PVDF and KB.

sintering condition and the result indicates that $Co_3O_4$ can react with $Li_2CO_3$ to generate $LiCoO_2$ under this condition (Fig. S5). Second, we designed an experiment of sintering $Li_2CO_3$-free LLZTO and $Co_3O_4$ under the same condition. The $La_2Zr_2O_7$ and $LiCoO_2$ were still found in the sintered products, indicating that lithium volatilization exists in LLZTO during the sintering process, and the volatilized lithium can react with $Co_3O_4$ to form $LiCoO_2$ (Fig. S6).

Based on the experimental results, Fig. 2b summarizes the process of the transformation. Initially, LLZTO exposed to air will form a $Li_2CO_3$ layer on the surface. Then, $Co_3O_4$ and LLZTO are fully mixed and sintered at 600 °C in air for 4 h. The $Li_2CO_3$ layer and $Co_3O_4$ undergo lithium-donor reaction to generate $LiCoO_2$ on the surface of LLZTO. Meanwhile, the lithium source inside LLZTO also reacts with $Co_3O_4$ to generate $LiCoO_2$ layer, but $Li_{6.4-3x}La_3Zr_{1.4}Ta_{0.6}O_{12}$ lithium-deficient phase is formed due to the loss of lithium, which contains many lithium defects and leads to the formation of $La_2Zr_2O_7$[17]. After that, in order to supply $La_2Zr_2O_7$ with lithium-ion and let it return to the original LLZTO structure, $Li_2O$ salt is added and sintered again at 600 °C for 5 h in air. Excitingly, the lithiumization of $La_2Zr_2O_7$ to LLZTO is realized. This process is the same as the preparation of LLZTO materials[18]. Finally, the pure LLZTO@LCO is obtained by washing and centrifuging. Different sintering temperatures (600, 700, 800, 900 °C) were attempted and we found that $LiCoO_2$ can be synthesized at all of the above temperatures. However, when the temperature is higher than 700 °C, the diffusion of elements occurs (Fig. S7), which is consistent with previously reported[19,20]. In addition, we also tried the one-step sintering of LLZTO@-$Li_2CO_3$, $Co_3O_4$, and $Li_2O$, but the structure of LLZTO coated with $LiCoO_2$ could not be formed and $Li_2CO_3$ still existed in store (Fig. S8).

**Stability and electrochemical activity of LLZTO@LCO.** LLZTO@LCO was then subjected to air-stability and activity measurements. To clarify its stability, the LLZTO@LCO was exposed to air for 4 months. The XRD results are shown in Fig. 3a. It is obvious that $Li_2CO_3$ was not formed after exposure to air for a long time. This means that the $LiCoO_2$ coating can restrain the formation of $Li_2CO_3$ layer and improve the stability of LLZTO significantly. The Fourier transform infrared (FTIR) spectra of the LLZTO and LLZTO@LCO samples exposed to air for 4 months also verified this result (Fig. 3b). It can be seen that the strong peaks of 1438 and 863 cm$^{-1}$ were formed in LLZTO, which corresponds to the $Li_2CO_3$ FTIR spectrum[9]. In addition, the weak peak of 3569 cm$^{-1}$ agrees with the LiOH H$_2$O FTIR spectrum, which is due to the reaction between water and LLZTO[9,10], but it does not affect the formation of $LiCoO_2$ layer (Fig. S9). Conversely, $Li_2CO_3$ and LiOH H$_2$O were not formed in LLZTO@LCO. To investigate lithium-intercalated activity of the LLZTO@LCO, it is used as the active material to prepare a cathode with a binder and conductive carbon, without the addition of any other active materials. A Li/LiClO$_4$-EC-DEC/ LLZTO@LCO liquid cell was then assembled (the inset in Fig. 3c). It should be noted that the liquid electrolyte is used here to describe the lithium-intercalated behavior of the LLZTO@LCO in more detail, so that it can be accurately compared with the characteristic redox peaks and charging/discharging voltage platforms (~3.9 V) of commercial $LiCoO_2$. As shown in Fig. 3c, charging and discharging voltage platforms are confirmed at about 3.9 V, which is corresponding to that of the commercial $LiCoO_2$. Figure 3d shows the cyclic voltammetry (CV) of the battery. There are strong redox peaks at about 3.87 and 3.95 V, indicating that the $LiCoO_2$ coating in LLZTO@LCO exhibits activity, improving the transport of ions on the LLZTO surface.

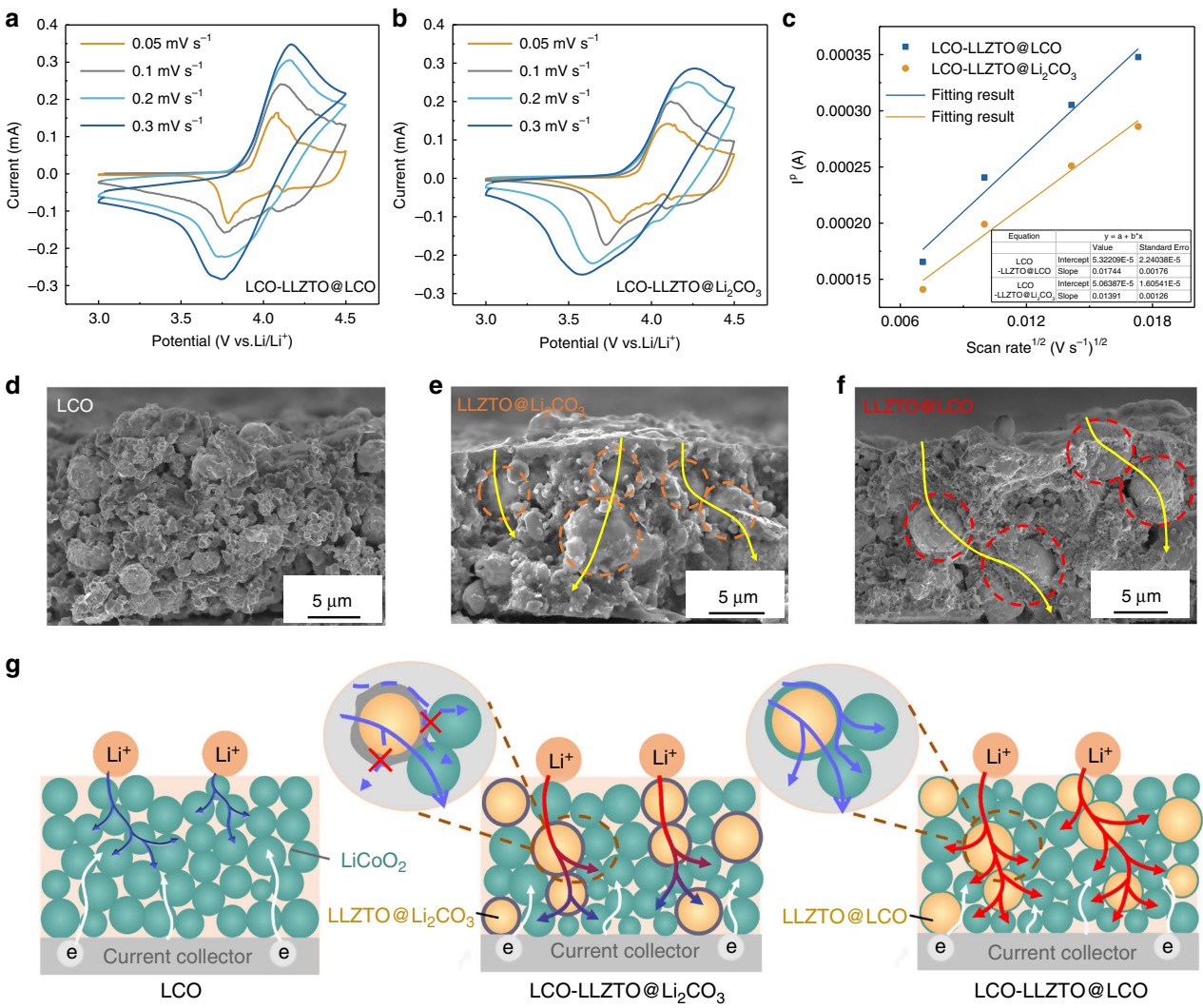

**Fig. 4 Characterization and ionic transport mechanism of the cathodes.** CV profiles of **a** LCO-LLZTO@LCO and **b** LCO-LLZTO@Li$_2$CO$_3$ cathodes at different scan rates. **c** Peak current as a function of the square root of the scan rate of LCO-LLZTO@LCO and LCO-LLZTO@Li$_2$CO$_3$ cathodes. **d–f** Cross-sectional SEM images of the LiCoO$_2$, LCO-LLZTO@Li$_2$CO$_3$ and LCO-LLZTO@LCO cathodes, separately. **g** Schematic illustration of the ionic transport mechanism inside cathodes.

Two weak peaks appear at 4.05 and 4.2 V, where LiCoO$_2$ lattice changes from hexagonal to monoclinic[21].

**Characterization and ionic transport mechanism of the LCO-LLZTO@LCO composite cathode.** In order to investigate the effect of LLZTO@LCO on the lithium-ion transfer kinetics of the cathode, the lithium-ion apparent diffusion coefficient was tested by performing CV measurements. Figure 4a, b shows the CV profiles of the LCO-LLZTO@LCO and LCO-LLZTO@Li$_2$CO$_3$ cathodes at different scan rates. The lithium-ion apparent diffusion coefficient can be calculated according to the Randles–Sevcik equation[22]

$$I_p = 2.68 \times 10^5 n^{3/2} A\, C\, D^{1/2} v^{1/2}, \tag{1}$$

where $I_p$ is the peak current (A); $n$ is the charge-transfer number of the redox reaction; $A$ is the area of the cathode plate (cm$^2$); $C$ is the lithium-ion concentration in LiCoO$_2$ cathode (0.051 mol cm$^{-3}$); $D$ is the lithium-ion diffusion coefficient (cm$^2$ s$^{-1}$); $v$ is the scan rate (V s$^{-1}$). $I_p$ is linearly related to $v^{1/2}$ and the value of $I_p/v^{1/2}$ can be obtained from the linear fitting results as 0.01744 and 0.01391 for LCO-LLZTO@LCO and LCO-LLZTO@Li$_2$CO$_3$ cathodes

(Fig. 4c). The lithium-ion apparent diffusion coefficient could be calculated to be $2.04 \times 10^{-13}$ cm$^2$ s$^{-1}$ for LCO-LLZTO@LCO cathode and $1.28 \times 10^{-13}$ cm$^2$ s$^{-1}$ for LCO-LLZTO@Li$_2$CO$_3$ cathode. Significantly, after Li$_2$CO$_3$ is converted to LiCoO$_2$, the lithium-ion diffusion coefficient of the cathode is increased by about 59%. This is because the activated LLZTO@LCO promotes ionic transport between particles, decreasing the resistance inside cathode. (Fig. S10).

To explore the mechanism by which LLZTO@LCO particles enhance the transport of lithium ions inside cathode, the cross-sectional micromorphology of the cathodes were observed. As shown in Fig. 4d, before adding LLZTO to the LiCoO$_2$ cathode, the LiCoO$_2$ nanoparticles are distributed on the cathode layer with a thickness of 15 µm, constructing a lithium-ion transport network. After LLZTO@Li$_2$CO$_3$ and LLZTO@LCO are introduced (Fig. 4e, f), they are distributed throughout the cathode and are in close contact with LiCoO$_2$ particles around, providing composite channels for lithium-ion transport. Significantly, LLZTO@Li$_2$CO$_3$ and LLZTO@LCO particles with a large particle size cross the cathode layer, which can construct rapid large-span channels for the transport of the lithium ions to the interior of the cathode. However, the Li$_2$CO$_3$ layer on the surface of LLZTO

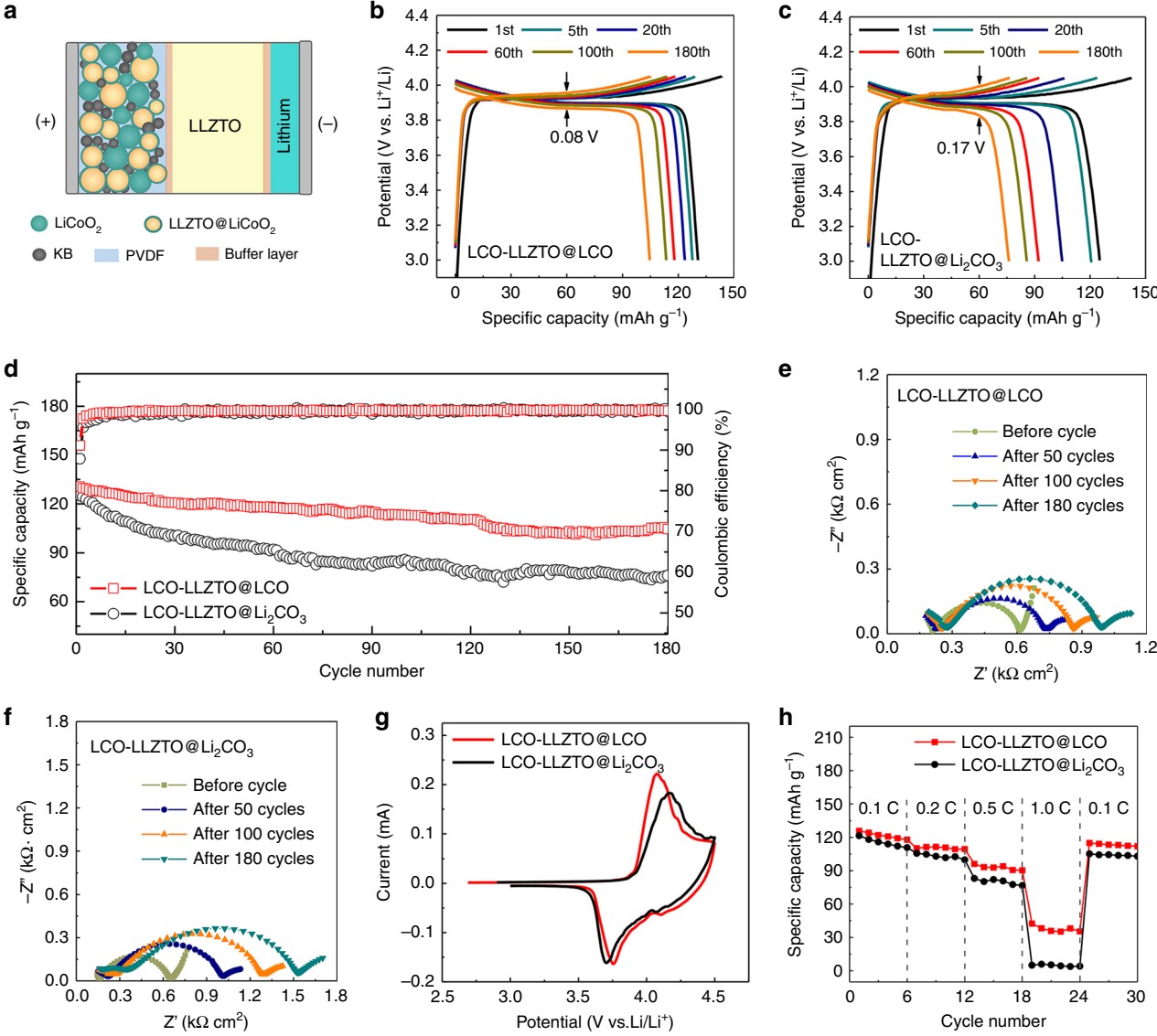

**Fig. 5 Electrochemical performance of the LLZTO-based SSBs. a** Schematic illustration of the LLZTO-based SSB. The buffer layer was formed by dissolving 10 wt% of lithium trifluoromethanesulfonyl (LiTFSI) in succinonitrile (SN), and polyacrylonitrile (PAN) was added to enhance film-forming property. **b**, **c** Discharge/charge curves of the SSBs with an LCO-LLZTO@LCO and LCO-LLZTO@Li$_2$CO$_3$ cathode, separately. **d** Cycling performance of the SSBs with an LCO-LLZTO@LCO and LCO-LLZTO@Li$_2$CO$_3$ cathode, separately. **e**, **f** EIS of the SSBs with a LCO-LLZTO@LCO and LCO-LLZTO@Li$_2$CO$_3$ cathode, separately. **g** CVs of the SSBs with an LCO-LLZTO@LCO and LCO-LLZTO@Li$_2$CO$_3$ cathode, separately. **h** Rate capability of the SSBs with an LCO-LLZTO@LCO and LCO-LLZTO@Li$_2$CO$_3$ cathode, separately. All tests were performed at room temperature.

with low conductivity will increase the interface impedance between the active material and the ion conductor particles. Conversely, LLZTO@LCO particles not only have an active surface in close contact with the active materials (Fig. S11), but also have a tight and low-impedance interface at the junction of LLZTO core and LiCoO$_2$ shell (Figs. S12 and 13), promoting the transport of lithium ions inside the cathode.

Based on the above results, it can be concluded that the optimization of the transport channels for lithium ions by LLZTO@LCO may be the reason for the improved ionic conductivity, thus a possible mechanism is provided in Fig. 4g. In the LiCoO$_2$ cathode without an ionic conductor, lithium ions diffuse into cathode through the ionic channels constructed by the active material LiCoO$_2$ with low ionic conductivity, which can reduce the diffusion rate and diffusion depth of lithium ions, causing a large voltage polarization and limiting electrode reaction to occur in the shallow layer of the cathode. But, after

LLZTO@Li$_2$CO$_3$ is added, large-span transport channels for lithium ions are formed around LLZTO@Li$_2$CO$_3$ particles, which can quickly transport lithium ions deeper into the LCO-LLZTO@Li$_2$CO$_3$ composite cathode. But, the presence of Li$_2$CO$_3$ on the surface of LLZTO is like sludge in the channels. Lithium ions can only enter and exit LLZTO particles from the thin layer of Li$_2$CO$_3$ and can only be transported to LiCoO$_2$ particles through the LLZTO bulk phase (Fig. S14), limiting the diffusion direction of lithium ions to the periphery, short of a crisscross lithium-ion transport network. Transforming Li$_2$CO$_3$ layer into active LiCoO$_2$ layer is like dredging the channels. Lithium ions can be freely transported not only in the bulk phase but also the surface of LLZTO@LCO, allowing the rapid lithium-ion transport paths to branch in any direction (Fig. S14), which realizes uniform diffusion of lithium ions on the shallow and deep layer of the LCO-LLZTO@LCO composite cathode. The rapid lithium-ion transport channel can be compared to an irrigation canal, in

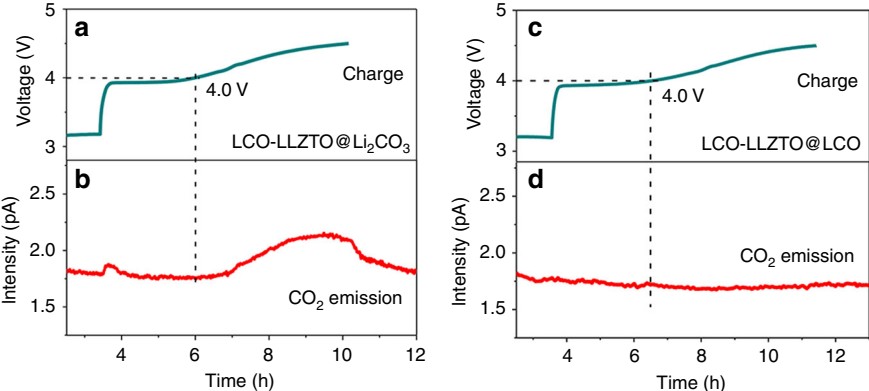

**Fig. 6 DEMS analysis of LLZTO@Li$_2$CO$_3$ and LLZTO@LCO in composite cathodes. a, b** Charge curves (top) and corresponding CO$_2$ emission (bottom) of LCO-LLZTO@Li$_2$CO$_3$ and LCO-LLZTO@LCO cathode, separately.

which the main channel is constructed along the high ionic conductivity area where more LLZTO@LCO particles are distributed, and the main channel branches to the surroundings to deliver lithium ions to various locations of the cathode, which greatly improves the transport efficiency of lithium ions. This transport mechanism allows lithium ions to be quickly and evenly distributed throughout the cathode, so the LCO-LLZTO@LCO composite cathode exhibits higher ionic conductivity.

**LLZTO-based SSBs with LCO-LLZTO@LCO composite cathode.** The electrochemical properties of the LLZTO@LCO and LLZTO@Li$_2$CO$_3$ were also compared in SSBs consisting of a lithium anode and LLZTO solid electrolyte pellet. The commercial LiCoO$_2$ was used as the active material, and LLZTO@Li$_2$CO$_3$ and LLZTO@LiCoO$_2$ were used as ionic conductors to prepare composite cathodes, separately. Then, the prepared LLZTO@-Li$_2$CO$_3$-containing LiCoO$_2$ (LCO-LLZTO@Li$_2$CO$_3$) cathodes and LLZTO@LCO-containing LiCoO$_2$ (LCO-LLZTO@LCO) cathodes were assembled into coin-type cells separately (Fig. 5a). Meanwhile, a thin buffer layer that is solid at room temperature was used to reduce the interface impedance between the electrolyte piece and the electrode plates (Fig. S15). All the cells were cycled at room temperature as well as 0.1 C (1 C = 140 mA g$^{-1}$). The cycling performance of the battery with an LCO-LLZTO@LCO cathode is shown in Fig. 5b. The discharge capacity of the first cycle reached 131 mA h g$^{-1}$ and the discharge capacity can be retained at 81% after 180 cycles with a voltage polarization of 0.08 V. Moreover, after 180 cycles the structure of LLZTO@LCO particles remained stable (Fig. S16). In contrast, the capacity retention of the battery with an LCO-LLZTO@Li$_2$CO$_3$ cathode reached only 60% after 180 cycles (Fig. 5c), but better than that of the battery with a pure LiCoO$_2$ cathode without an LLZTO@LCO or LLZTO@Li$_2$CO$_3$ ionic conductor (Fig. S17). Furthermore, the CE of the battery with an LCO-LLZTO@LCO cathode reaches 91.1% at first cycle and stable at above 99% after the first five cycles, but the battery with an LCO-LLZTO@Li$_2$CO$_3$ cathode exhibited lower CE of 87.8% at first cycle (Fig. 5d). The increase in CE is owing to the reduction of side reactions by removing the Li$_2$CO$_3$ inside the cathode. Figure 5e, f shows the electrochemical impedance spectroscopy of the batteries. It can be seen that the impedance plot includes an incomplete semicircle in the high frequency region, a semicircle in the middle frequency region and a tail in low frequency region, in which the semicircle in the middle frequency region corresponds to the overall interface resistance ($R_{int}$) inside the battery. The interface $R_{int}$ in the battery with an LCO-LLZTO@LCO cathode is 600 Ω cm$^2$ after 180 cycles, lower than 988 Ω cm$^2$ in the battery with an

LCO-LLZTO@Li$_2$CO$_3$ cathode (Fig. S18). The smaller interface resistance is mainly due to the optimized ion transfer channels of the LLZTO@LCO particles. In addition, the CVs measured on the composite cathodes (Fig. 5g) show that LCO-LLZTO@LCO cathode has a lower polarization. This can be explained by the fact that the transformation of insulating Li$_2$CO$_3$ to active LiCoO$_2$ with high ionic conductivity achieves interface homogeneity inside cathode and can speed up the transport of lithium-ion between the particles. The change of LiCoO$_2$ lattice from hexagonal to monoclinic is also observed at about 4.05 and 4.2 V by testing d$Q$ d$V^{-1}$ of the SSB with an LCO-LLZTO@LCO cathode (Fig. S19), but that is not obviously shown in the CV curve in Fig. 5g. This is because the SSB has a higher impedance than the liquid battery, leading to a larger polarization, which results in a shift and widen in the main peak of the LiCoO$_2$ so that the weak peaks at 4.05 and 4.2 V are covered. Figure 5h shows the rate performance of the batteries. The discharge capacity of the battery with an LCO-LLZTO@LCO cathode still reached 116 mA h g$^{-1}$ at 0.2 C and 100 mA h g$^{-1}$ at 0.5 C, but the discharge capacity of the battery with an LCO-LLZTO@Li$_2$CO$_3$ cathode only reached 105 and 80 mA h g$^{-1}$ at 0.2 and 0.5 C, which corresponds to higher ionic conduction of the LCO-LLZTO@LCO cathode. Significantly, instead of using low-voltage active materials which were mostly used in the LLZO-based SSBs in previous reports, the high-voltage LiCoO$_2$ active materials with LLZTO@LCO ionic conductor are used to prepare composite cathodes to assemble solid cells in this work, and show improved cycleability and rate performance (Fig. S20).

**Discussion**

Purity of electrolytes has guided the history of commercial batteries. For instance, the successful development of high-purity LiPF$_6$ in 1994, coupled with 99.9% pure ethylene carbonate, offered a leap forward in cycling ability of commercial lithium-ion batteries. The ubiquitous Li$_2$CO$_3$ can be considered as an impurity of LLZO particles, leading to inferior purity (<95%), which is far away from the practical needs. Hence, when using in LiCoO$_2$ cathodes, the purity of the LLZO electrolyte is equivalent to 100% owing to the substitution of insulating Li$_2$CO$_3$ impurity for active LiCoO$_2$, and the latter provides a homogeneous contact with the LiCoO$_2$ active material inside composite cathodes. This is an apparent advantage of the LLZTO@LCO from the view point of electrolyte purity.

To clarify the electrochemical difference of LLZTO@Li$_2$CO$_3$ and LLZTO@LCO inside the composite cathode, the LCO-LLZTO@Li$_2$CO$_3$ and LCO-LLZTO@LCO composite cathodes were analyzed by the differential electrochemical mass

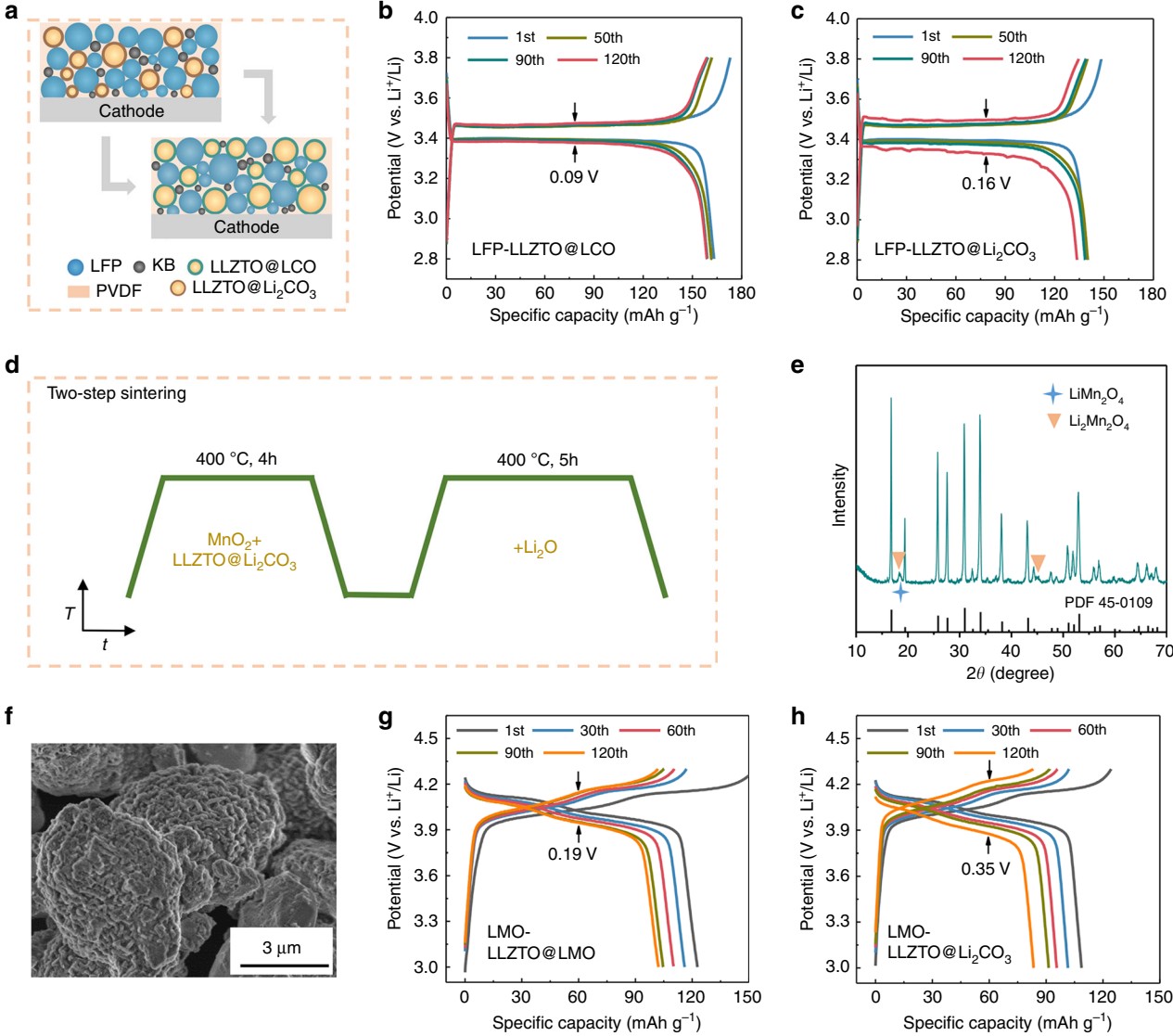

**Fig. 7 Extensive applicability of LLZTO@active-material and the two-step solid state reaction. a** Schematic illustration of the LFP-LLZTO@Li$_2$CO$_3$ and LFP-LLZTO@LCO composite cathodes. **b**, **c** Discharge/charge curves of the SSBs with an LFP-LLZTO@LCO and LFP-LLZTO@Li$_2$CO$_3$ composite cathode, separately. **d** Schematic illustration of the two-step solid state reaction. **e**, **f** SEM image and XRD pattern of the ultimate materials after converting Li$_2$CO$_3$ into LMO. **g**, **h** Discharge/charge curves of the SSBs with an LMO-LLZTO@LMO and LMO-LLZTO@Li$_2$CO$_3$ composite cathode, separately.

spectrometry analysis (DEMS) separately. The assembled liquid batteries were used to detect the release of CO$_2$. Figure 6a shows the charge curve (top) and corresponding CO$_2$ emission (bottom) for the LCO-LLZTO@Li$_2$CO$_3$ composite cathode. Notably, the intensity of CO$_2$ began to increase when charged to about 4.0 V (vs. Li/Li$^+$), which is considered to be due to the decomposition of Li$_2$CO$_3$ in LLZTO@Li$_2$CO$_3$, consistent with previous observations that Li$_2$CO$_3$ was decomposed above 4.0 V[23,24]. It should be noted that this is the first demonstration of the electrochemical decomposition of Li$_2$CO$_3$ formed on the surface of LLZTO by experiments. In stark contrast, benefiting from the transformation of Li$_2$CO$_3$ into LiCoO$_2$, CO$_2$ was not released in the homogeneous LCO-LLZTO@LCO composite cathode (Fig. 6b), exhibiting much higher electrochemical stability, which explains the high initial CE of the battery with an LCO-LLZTO@LCO composite cathode.

We further evaluated the activity and extensive applicability of LLZTO@LCO by adding it to LiFePO$_4$ cathodes with a low ionic diffusion rate. The prepared LLZTO@LCO-containing LiFePO$_4$ (LFP-LLZTO@LCO) composite cathode and LLZTO@Li$_2$CO$_3$-containing LiFePO$_4$ (LFP-LLZTO@Li$_2$CO$_3$) composite cathode (Fig. 7a) were assembled into SSBs using a lithium anode and a LLZTO solid electrolyte pellet, separately. As shown in Fig. 7b, the initial discharge capacity of the battery with an LFP-LLZTO@LCO cathode reached 163.2 mA h g$^{-1}$, closing to the theoretical capacity of 170 mA g$^{-1}$, and can be retained at 97% after 120 cycles with a low-voltage polarization of 0.09 V. In contrast, the battery with an LFP-LLZTO@Li$_2$CO$_3$ cathode exhibits an initial discharge capacity of 146.4 mA h g$^{-1}$ and has a large voltage polarization of 0.16 V after 120 cycles (Fig. 7c). In addition, the SSB with an LFP-LLZTO@LCO cathode exhibits longer-term cycling performance than that previously reported (Fig. S21)[4,25–29].

The two-step solid-state reaction process (Fig. 7d) was also successfully extended to prepare LLZTO@LiMn$_2$O$_4$. Figure 7e shows the XRD pattern of the converted materials. The ultimate materials contain LLZTO, LiMn$_2$O$_4$, and Li$_2$Mn$_2$O$_4$, in which the Li$_2$Mn$_2$O$_4$ is a discharged state of LiMn$_2$O$_4$ due to the presence of excess lithium salt during sintering, causing the insertion of lithium ions into LiMn$_2$O$_4$. Figure 7f shows the SEM image of LLZTO@LiMn$_2$O$_4$/Li$_2$Mn$_2$O$_4$ (LLZTO@LMO). LiMn$_2$O$_4$ and

$Li_2Mn_2O_4$ with a nanoparticle-like morphology are evenly distributed on the surface of LLZTO to form a coating. The electrochemical properties of the LLZTO@LMO-containing $LiMn_2O_4$ (LMO-LLZTO@LMO) and LLZTO@$Li_2CO_3$-containing $LiMn_2O_4$ (LMO-LLZTO@$Li_2CO_3$) composite cathodes were compared in SSBs with the lithium anode and LLZTO electrolyte pellet. The initial discharge capacity of the battery with an LMO-LLZTO@LMO cathode reached 122.8 mA h g$^{-1}$, which is higher than that of another battery (108.7 mA h g$^{-1}$). In addition, compared with the battery with an LMO-LLZTO@$Li_2CO_3$ cathode, the battery with an LMO-LLZTO@LMO cathode exhibits improved cycling stability (Fig. S22) and a lower voltage polarization of 0.19 V after 120 cycles at 0.1 C (1 C = 148 mA g$^{-1}$), while the initial CE is lower due to the presence of discharged $Li_2Mn_2O_4$ (Fig. 7g, h).

In conclusion, by transforming the ubiquitous insulating $Li_2CO_3$ layer on the surface of LLZTO solid electrolytes into an active $LiCoO_2$ layer, pure LLZTO@LCO particles are successfully synthesized with an on-surface lithium-donor reaction. The R-3m symmetry $LiCoO_2$ is mainly generated by the lithium donor reaction of the $Li_2CO_3$ layer and $Co_3O_4$ in the first sintering. At the same time, it is found that lithium volatiles from LLZTO also reacts with $Co_3O_4$, resulting in part of $LiCoO_2$, accompanying by $La_2Zr_2O_7$ impurity. At the heart of our technology is offsetting the formidable impurity $La_2Zr_2O_7$ by precisely supplementing extra lithium sources, thus restoring it to the pristine LLZTO structure in the second sintering step. The LLZTO@LCO particles are exposed to air for 4 months without $Li_2CO_3$ formation, indicating excellent store stability and demonstrating a radical solution of the $Li_2CO_3$ issue. We found that the decomposition of $Li_2CO_3$ formed on the surface of LLZTO occurs at voltages above 4.0 V, which is one of the reasons for the low initial Coulomb efficiency. Meanwhile, the converted $LiCoO_2$ layer of the LLZTO@LCO particles exhibits the same lithium-intercalated electrochemical activity with commercial $LiCoO_2$, which enables it to interact favorably with the $LiCoO_2$ active material in the solid LCO-LLZTO@LCO composite cathode. As a consequence of the interface homogeneity inside cathode, the solid-state lithium metal battery with the LLZTO@LCO and $LiCoO_2$ composite cathode shows 81% capacity retention after 180 cycles at 0.1 C, room temperature, superior to that with the LLZTO@$Li_2CO_3$ and $LiCoO_2$ one, representing the highest level among $LiCoO_2$-based solid batteries. Our results indicate that although the formation of $Li_2CO_3$ on LLZO is inevitable, it would no longer hinder. Lithium-ion transfer at the solid electrolyte/cathode interface, but provide a chance to be transformed into active materials, thus achieving an in-situ intimate contact of ion conductor and active materials inside the cathode. In addition, the solid sintering reaction, which is the most common mass production method for ceramics-type electrolytes and cathode materials, has also been successfully applied to the in-situ transformation of $Li_2CO_3$ to $LiMn_2O_4$. It is hopeful to develop a series of LLZO@$LiFePO_4$, LLZO@layered Ni-Co-Mn or Ni-Co-Al, etc. to precisely match active materials inside the composite cathodes for solid-state lithium metal batteries.

## Methods

**LLZTO@LiCoO₂ materials**. LLZTO@$LiCoO_2$ materials were prepared by a two-step solid-state reaction process. LLZTO powders and $Co_3O_4$ (Aladdin, 99.99%) were mixed in a mass ratio of 20:3 at an agate mortar for 10 min and sintered at 600 °C for 4 h. Then $Li_2O$ were added to the precursors in a mass ratio of 3:10, and heated to 600 °C and dwelled on for 5 h. The obtained materials were finally washed with ethyl alcohol and centrifuged giving rise to LLZTO@$LiCoO_2$. The LLZTO powders and pellets was prepared by a method previously reported[30,31].

**Composite cathodes**. The LCO-LLZTO@LCO and LCO-LLZTO@$Li_2CO_3$ composite cathodes were prepared in the air. LLZTO@$LiCoO_2$ and LLZTO@$Li_2CO_3$ solid electrolytes were, respectively, mixed with $LiCoO_2$ active materials, PVDF, KB

in a mass ratio of 3:5:1:1 in N-methylpyrrolidone (NMP) solvent. After stirring for 12 h, the slurry was scraped on the carbon-containing aluminum foil, and heated at 60 °C in atmospheric pressure for 2 h, then dried at 80 °C in vacuum for 24 h to obtain cathode foil and cut it into discs of 12 mm in diameter. A total of 80 wt% of commercial $LiCoO_2$, 10 wt% of PVDF, and 10 wt% of KB were mixed to prepare pure $LiCoO_2$ cathodes. The LFP-LLZTO@LCO and LFP- LLZTO@$Li_2CO_3$ composite cathodes were prepared by the same method as above.

**Assembly of liquid cells**. CR2032-type liquid coin cells were assembled in an argon-filled glovebox to detect the air-stability and activity of LLZTO@LCO. The cathodes were prepared by mixing LLZTO@$LiCoO_2$ active materials, PVDF and KB in a mass ratio of 8:1:1 in NMP solvent. After stirring for 12 h, the slurry was scraped on the carbon-containing aluminum foil, and heated at 60 °C in atmospheric pressure for 2 h, then dried at 80 °C in vacuum for 24 h to obtain cathode plate and cut it into discs of 12 mm in diameter. Li foil with 12 mm in diameter was used as anode and dissolving 0.1 M $LiClO_4$ in EC-DEC (1:1, v/v) was used as a liquid electrolyte.

**Assembly of solid-state cells**. CR2032-type solid-state coin cells were assembled in an argon-filled glovebox. In all-solid-state cells, the LLZTO plates were used as SSEs and Li foils with 12 mm in diameter were used as anodes. In order to improve the interface between SSE and electrodes, a buffer layer that exhibits a film at room temperature was introduced, which was formed by dissolving 10 wt% of tri-fluoromethanesulfonyl in succinonitrile at 80 °C and adding polyacrylonitrile to enhance film-forming property. The gelatinous slurry was scraped on the electrode surface at 80 °C, and cooled down to room temperature to form a solid film.

**Electrochemical measurement**. The charge/discharge tests of the cells were carried out using Land machines at room temperature. The specific capacity of the batteries with an LCO-LLZTO@LCO cathode was calculated based on the weight of the cathode active materials including both the $LiCoO_2$ on the surface of LLZTO@$LiCoO_2$ and the commercial $LiCoO_2$. The proportion of Co element is 11.58 wt% in the LLZTO@$LiCoO_2$, which is provided by the inductively coupled plasma spectrum test. The loading of cathodes is about 2 mg cm$^{-2}$, corresponding to the active materials of around 1.12 mg cm$^{-2}$.

**TEM, selected area electron diffractions, and high resolution transmission electron microscopy observation**. The coating structure of LLZTO@$Li_2CO_3$ and LLZTO@$LiCoO_2$ were observed using Field Emission JEM-2100F TEM. The diffraction fringes and lattice fringes of LLZTO@$Li_2CO_3$ and LLZTO@$LiCoO_2$ were observed using selected area electron diffractions and high resolution transmission electron microscopy of field emission JEM-2100F TEM, separately.

**SEM observation and energy-dispersive X-ray spectroscopy analysis**. The microstructures and X-ray (energy-dispersive X-ray spectroscopy) mapping of LLZTO and LLZTO@$LiCoO_2$ were observed using SU-8220 field emission SEM.

**X-ray powder diffraction and FTIR analysis**. The phases and crystalline structure of the materials before and after transformation were analyzed using X-ray powder diffraction with Cu Kα radiation. LLZTO and LLZTO@$LiCoO_2$ were exposed to air for four months for FTIR measurements.

**Differential electrochemical mass spectrometry analysis**. The LCO-LLZTO@LCO and LCO-LLZTO@$Li_2CO_3$ slurry were dripped on stainless steel with 12 mm in diameter, separately. Then, the dried composite cathodes were assembled into customized Swagelok cells. Dissolving 1 M $LiPF_6$ in EC/DMC (1:1 v:v) was used as the liquid electrolyte and Li foil with 12 mm in diameter was used as the anode. The liquid cells were charged to 4.5 V at 0.25 C.

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

## Acknowledgements

This work was supported financially by the National Natural Science Foundation of China under grant No. 51672299.

## Author contributions

T.Z. and Y.-N.Y. conceived and designed the experiments. Y.-N.Y., and Y.-X.L. performed the experiment. T.Z. and Y.-N.Y. carried out the data analysis, discussed, and wrote the paper. Y.-Q.L. prepared LLZTO powders and pellets.

## Competing interests

The authors declare no competing interests.
