## [Peer Review File · Nature Communications]

Reviewers' Comments:

Reviewer #1 (Remarks to the Author):

This paper reports a method to remove the Li_2CO_3 on the LLZTO surface by forming a LiCoO_2 layer, which might help improve the preparation of composite cathode in garnet-based solid-state batteries. However, it is unsuitable for publication due to the following concerns:

1. How did the authors make sure that the whole LLZTO particles have been covered by Co_3O_4 , by simple mixing? What is the mechanism?
2. The biggest concern is the preparation of cathode for the "solid-state cells" as shown in Fig. 4a. How did Li^+ transport between LCO and LLZO@LCO particles in the composite cathode? What is the function of the LLZO in the cathode with gel electrolyte used, to increase the ionic conductivity? This puts Fig. 4 and Fig. 5 into questions. As described in the Experiment, the gel "was dripped on the electrode surface", how much gel electrolyte was used? How deep it can penetrate? Comparison between LLZO and LCO-LLZO@LCO cathode with the gel electrolyte should be made. Moreover, an all-ceramic composite cathode by co-sintering is strongly suggested to demonstrate the advantage of LLZO@LCO particles.

Reviewer #2 (Remarks to the Author):

This study attempts to address the issue of high interface resistance between cathode and LLZO solid electrolyte. Their approach is to introduce a layer of Li_2CO_3 or allow the growth of native Li_2CO_3 to grow on the surface of particles to act as a reactive layer during heating. Unfortunately, the lack of clarity of the concept prevents comprehension of what is proposed. Below are comments.

- 1) The use of the term peritectic and peritectoid is not correct. There is no gas byproduct in these reactions.
- 2) Indeed Li_2CO_3 is not good for the Li interface, but it literature did not necessarily imply it 's bad for forming cathode interfaces.
- 3) The biggest concert is the scientific process and presentation of results:
 - a. What is the science here? Is it how the layer or Li_2CO_3 enables a high performance interface between LCO and LLZTO? If so, there is no SEM data on the microstructure showing what the electrodes look like, There is not electrochemical data that measures the interface resistance.
 - b. The most concerning is the lack of clarity on whether or not a liquid electrolyte is used in the electrochemical data? Figure 4 uses liquid electrolyte to characterize the behavior of the cathode. Figure 5 uses LiTFSI and succinonitrile. Why the change in electrolytes?
 - c. Why was a liquid electrolyte used at all? The way the introduction was written, the impression is given that the goal is to eliminate the use of liquid electrolyte in the cathode by reducing interface resistance.

d. Where is the interface resistance in the EIS in Figure 5? I would think the main reason for doing EIS is to measure the interface resistance to validate that the approach works.

4) Overall, the authors need to be clear about how the quality of the solid-solid interface affects cell performance of all solid-state cells.

Reviewer #3 (Remarks to the Author):

Lithium garnets are promising electrolytes to enable the next-generation all-solid-state lithium batteries with reliable safety, however, the spontaneous formation of Li_2CO_3 on the surface of LLZO is always considered as an inherent drawback of this promising material. This paper reports an innovative method by reacting the Li_2CO_3 layer with CoO_2 to in-situ form LiCoO_2 cathode. The in-situ formed LiCoO_2 not only ensures a great interfacial contact with LLZO electrolyte but also prevent further reaction of LLZO with $\text{H}_2\text{O}/\text{CO}_2$ through quantitatively monitoring the CO_2 emission at different voltages. From the interphase transition point of view, this work represents an important achievement which indicates that interface homogeneity is the key to increasing the electrochemical performance of solid-state lithium batteries. The results are interesting and provide novel insights to address the interfacial challenges in garnet-based batteries. Therefore, it is recommended to be accepted in Nature Communications.

(1) According to Fig. 2b, the lithium carbonate layer and cobalt oxide undergo the on-surface peritectoid reaction to produce lithium cobalt oxide. At the same time, the lithium source inside LLZTO also reacts slightly with the cobalt oxide to generate the lithium cobalt oxide layer. In these two reactions, which one dominates the lithium cobalt oxide layer?

(2) Figure 5 shows the DEMS results of carbon dioxide emission. The cells were assembled by lithium cobalt oxide active materials, mixed with LLZTO electrolytes with and without lithium cobalt oxide layer, separately. How about that of the commercial lithium cobalt oxide active materials without the LLZTO electrolyte as the ionic conductor?

(3) The buffer layer does not use a liquid solvent or a drying process. What is the state of the buffer layer in the solid-state lithium battery? Detailed information should be provided.

(4) Typos. Page 2, line 18: maintains; Page 2, line 21: the intimate; Page 3, line 35, in the recent year; Page 6, line 103: after the first sintering.

Author's Response to Reviewer #1

1. How did the authors make sure that the whole LLZTO particles have been covered by Co_3O_4 , by simple mixing? What is the mechanism?

Response: Thank you very much for your comments. Herein, the distribution of Co_3O_4 in LLZTO and the sintering mechanism of the LiCoO_2 layer on the surface of LLZTO are explained based on geometric calculation, SEM results, and previous reports.

To make sure that Co_3O_4 can be widely distributed in LLZTO powders and fully react with Li_2CO_3 on the surface of LLZTO, we estimated the mass ratio of Li_2CO_3 in the LLZTO before the experiment and used an excessive amount of Co_3O_4 to mix and sinter. According to the SEM image of the LLZTO particles in the manuscript, we know that the diameter of the LLZTO particle is about $4\ \mu\text{m}$, and the thickness of the Li_2CO_3 layer is about $150\ \text{nm}$ on average. The density of Li_2CO_3 and LLZTO is 2.11 and $5.5\ \text{g cm}^{-3}$ respectively. Assuming the LLZTO particle is a sphere, the molar ratio of Li_2CO_3 to LLZTO can be calculated as 1: 2. Meanwhile, the molar ratio of Co_3O_4 to LLZTO is 1: 2 (the mass ratio is 20:3) in the mixed raw materials, hence the molar ratio of Li_2CO_3 , Co_3O_4 and LLZTO is 1: 1: 2. According to Equation (1), we can know that Co_3O_4 is excessive in the transformation reaction, which is beneficial to the wide distribution of Co_3O_4 in the mixture to fully eliminate lithium carbonate.

Additionally, Fig. 1 shows the SEM images of the $\text{Co}_3\text{O}_4/\text{LLZTO}$ mixture after grinding. The mixture was compressed into a tablet before testing for easy observation. From Fig. 1a, it can be seen that the LLZTO particles are buried by Co_3O_4 fine particles, similar to rocks in sand, realizing that whole LLZTO particles are uniformly covered by Co_3O_4 , by simple mixing. At the same time, the Co_3O_4 particles adhere to the entire surface of the LLZTO particle (Fig.1b, c), which may be related to the smaller particle size and hardness of Co_3O_4 compared to LLZTO.

Fig. 1 SEM images of the $\text{Co}_3\text{O}_4/\text{LLZTO}$ mixture at different magnifications.

Fig. 2 Schematic illustration of the mechanism of the solid sintering reaction

The sintering mechanism is provided. A solid-state reaction generally includes three processes: chemical reaction on the phase interface, mass migration, and crystal growth. The contact of the reactants is a prerequisite for chemical reaction and the sintering temperature above the Tammann temperature (that is at approximately half the temperature of the melting point (in K)) is a necessary condition for obvious diffusion inside the system.¹ As shown in Fig. 2, during the sintering process, the reaction begins at the interface between the Li_2CO_3 layer and Co_3O_4 to generate LiCoO_2 . When the generated LiCoO_2 reaches a certain thickness at the interface and the sintering temperature is higher than the Tammann temperature ($\text{Co}_3\text{O}_4 \sim 311^\circ\text{C}$, $\text{Li}_2\text{CO}_3 \sim 294^\circ\text{C}$), the constituent ions of reactants diffuse to each other through the lattice, surface, grain boundary or dislocation of the LiCoO_2 and react inside the reactants, which leads to the extension of the LiCoO_2 interface layer. The LiCoO_2 layers at different reaction sites extend simultaneously and contact with each other, achieving a complete coating layer on the entire surface of LLZTO particle. Subsequently, as the grain grows, a dense LiCoO_2 coating is formed (Fig. 3).

Fig. 3 SEM image of the LLZTO@LCO particle

2. The biggest concern is the preparation of cathode for the “solid-state cells” as shown in Fig. 4a (Fig. 5a in the revised manuscript). How did Li^+ transport between LCO and LLZO@LCO particles in the composite cathode? What is the function of the LLZO in the cathode with gel electrolyte used, to increase the ionic conductivity?

Response: As reviewer’s states, it is necessary to consider the function of the LLZO in the cathode with buffer layer used. In order to clarify the function of LLZO inside cathode, we first exclude the possibility that the buffer layer can penetrate into the cathode. As shown in Fig. 4a, the buffer layer is semi-fluid and has strong viscoelasticity at 80 °C, which makes it difficult to diffuse into the cathode. Meanwhile, it is solid at room temperature, and has high elasticity as well as can be processed into a film (Fig. 4b), which ensures the solid state structure of the assembled battery at room temperature. Additionally, there is a clear boundary between the ultra-thin buffer layer and the cathode (Fig. 4c). The energy dispersive X-ray (EDX) mapping analysis results that the N elements of SCN are only distributed on the buffer layer area and are clearly layered with the Co elements of the cathode layer (Fig. 4d), further confirming that the buffer layer has no effect on the internal structure of the cathode. Therefore, in the battery structure shown in Fig. 5a in the manuscript, the ionic conductivity inside the composite cathode is only contributed by the LLZO particles and active material LiCoO_2 . The buffer layer does not penetrate into the cathode.

Fig. 4 Characterization of the buffer layer. a, b Optical photos of the buffer layer at 80 °C and room temperature. c Cross-sectional SEM image of the LCO-LLZTO@LCO composite cathode coated with buffer layer. The thickness of the buffer layer at the interface cannot be recognized due to its very thinness. d EDX analysis of the LCO-LLZTO@LCO composite cathode coated with

buffer layer. (This figure is used as Fig. S13 in supplementary information)

Directly, the ionic conductivity of LCO, LCO-LLZTO@LCO and LCO-LLZTO@Li₂CO₃ cathodes were measured by electrochemical impedance spectroscopy (EIS), respectively, to demonstrate the increase of the cathode's ionic conductivity by adding LLZTO particles. It should be noted that the electronic conductor of Ketjen Black (KB) is not added when measuring the ionic conductivity. The cathode slurry without KB was coated on the carbon-containing aluminum foil (as the blocking electrode) with a diameter of 15.5 mm, and another piece of aluminum foil was pressed on the surface of the cathode after semi-drying. Then the Al/Cathode without KB/Al sandwich structure samples were dried under vacuum. The a.c. impedance was measured in the frequency range of 1 MHz to 1 Hz.

Fig. 5a shows the impedance plots of Al/Cathode without KB/Al. The cathodes exhibit an impedance behavior similar to the solid electrolyte, including an incomplete semicircle at a high frequency of 1 MHz and a slanted tail at low frequency,^{2,3} which indicates that the KB-free cathodes mainly have ionic conductivity rather than electronic conductivity. This is because the cathodes do not contain the KB that dominates the electronic conduction, but the LiCoO₂ active material and the LLZTO ion conductor which dominate ionic conduction. The intercepts on the resistance axis are the ionic resistance (R_i) of the cathodes. The ionic conductivity of the cathodes were calculated using $\sigma = \frac{1}{R} \frac{L}{A}$, where σ is the ionic conductivity of the cathode; R is the ionic resistance R_i ; L and A are the cathode thickness and measurement area, respectively. The calculated results of the ionic conductivity are shown with a blue line in Fig. 5b. The LCO-LLZTO@LCO composite cathode exhibits the highest ionic conductivity of $5.52 \times 10^{-5} \text{ S cm}^{-1}$, improving the ionic conductivity of the pure LiCoO₂ cathode by nearly 650%, which implies that the LLZTO@LCO ionic conductor accelerates the transport of lithium ions to the other side of the cathode. Simultaneously, the activated LLZTO surface obtained by converting Li₂CO₃ to LiCoO₂ reduces the ionic transport resistance at the interface between the LiCoO₂ and LLZTO particles inside cathode. As a result, the ionic conductivity of the LCO-LLZTO@LCO composite cathode is further improved compared with that of the LCO-LLZTO@Li₂CO₃ composite cathode ($1.74 \times 10^{-5} \text{ S cm}^{-1}$).

Fig. 5 Measurement of the conductivity of the cathodes. a EIS of the Al/Cathode without KB/Al. **b** The ionic and electronic conductivity of the cathodes. **c** The impedance plot of the case that both ionic and electronic transport are significant.⁵ **d** EIS of the Al/Cathode with KB/Al. (This figure is used as Fig. S10 in supplementary information, and parts of this figure are used as Fig. 4 in the manuscript)

Additionally, the Al/Cathode with KB/Al was used to investigate the effect of LLZTO particles on the electron transport behavior of the cathode. It has been reported that when a mixed conductor has both significant ionic and electron transport, its impedance plot using blocking electrodes is shown as Fig. 5c, and the equivalent circuit consists of an electronic current path and an ionic current path in parallel (the inset in Fig. 5c). C_{geom} is the parallel plate capacitance of the two electrodes and C_{int} is the interface capacitance between electrodes and the intermediate medium. R_e is the electronic resistance, represented by the intercept (R_1) on the resistance axis at low frequencies. R_i is the ionic resistance, which in parallel with R_i is presented by the intercept (R_2) on the resistance axis at high frequencies. On the other hand, when the electronic conduction is dominant, the electron current will be much larger than the ionic current, resulting in a short circuit of the interface capacitance C_{int} , thus only presenting a semicircle in the impedance plot. Meanwhile, no capacitive tail appears at low frequency due to the shunt of the electronic current path.^{4,5} Therefore, as shown in Fig. 5d,

the semicircles without capacitive tail in the impedance plots of the Al/Cathode with KB/Al indicate that the cathodes with KB are the mixed conductors whose electronic resistance is much lower than the ionic resistance. This can be ascribed to the fact that KB plays a major role in electronic conduction inside cathode. The electronic conductivity of the cathodes with KB were also calculated using $\sigma = \frac{1}{R} \frac{L}{A}$, where σ is the electronic conductivity of the cathode; R is the electronic resistance R_e , obtained from the intercept on the resistance axis; L and A are the cathode thickness and measurement area, respectively. The results of the calculated electronic conductivity are shown with a yellow line in Fig. 5b. It can be seen that the electronic conductivity of the LCO, LCO-LLZTO@LCO and LCO-LLZTO@Li₂CO₃ cathodes are almost the same ($\sim 1 \times 10^{-3}$ S cm⁻¹). Combined all of the above EIS results with and without KB, we can draw a conclusion that in the composite cathode, LLZTO@LCO plays a major role in enhancing ionic-conduction, but has almost no effect on electron-conduction.

Fig. 6 Cross-sectional SEM images of the cathodes. a LiCoO₂ cathode. **b** LCO-LLZTO@Li₂CO₃ composite cathode. **c** LCO-LLZTO@LCO composite cathode. **d** LCO-LLZTO@LCO composite cathode at high magnification. (Fig. 6a-c are used as Fig. 4d-f in the manuscript. Fig. 6d is used as Fig. S11 in supplementary information)

In order to explore the mechanism by which LLZTO@LCO particles enhance the transport of lithium ions inside cathode, the cross-sectional micromorphology of the cathodes were observed. As shown in Fig. 6a, before adding LLZTO to the LiCoO₂ cathode, the LiCoO₂ nanoparticles are distributed on the cathode layer with a thickness of 15 μm, constructing a lithium-ion transport

network. When $\text{LLZTO@Li}_2\text{CO}_3$ is added, the $\text{LLZTO@Li}_2\text{CO}_3$ particles are distributed throughout the cathode and are in close contact with LiCoO_2 nanoparticles around (Fig. 6b), providing composite channels for lithium-ion transport. Significantly, as can be seen from Fig. 6b and 6c, after the $\text{LLZTO@Li}_2\text{CO}_3$ and LLZTO@LCO are introduced to the cathodes, separately, they have large particle size and cross the cathode layer, which can construct rapid large-span channels for the transport of the lithium ions to the interior of the cathode. However, the Li_2CO_3 layer on the surface of LLZTO has low conductivity, which increases the interface impedance between the active material and the ion conductor particles. Conversely, LLZTO@LCO particles have an active surface in close contact with the active materials (Fig. 6d), which is more conducive to the diffusion of lithium ions between LLZTO ionic conductor and LiCoO_2 active material.

Fig. 7 Schematic illustration of the ionic transport mechanism inside cathode. (This figure is used as Fig. 4g in the manuscript)

Based on the above results, it can be concluded that the optimization of the transport channels for lithium ions by LLZTO@LCO is the reason for the improved ionic conductivity. A speculated mechanism is provided in Fig. 7. In the pure LiCoO_2 cathode without an ionic conductor, lithium ions diffuse into the interior of cathode through the ionic channels constructed by the active material LiCoO_2 with low ionic conductivity, which reduces the diffusion rate and diffusion depth of lithium ions, causing a large voltage polarization and limiting electrode reaction to occur in the shallow layer of the cathode. But, after $\text{LLZTO@Li}_2\text{CO}_3$ is added, large-span transport channels for lithium ions are formed around $\text{LLZTO@Li}_2\text{CO}_3$ particles, which can quickly transport lithium ions deeper into the cathode through the LLZTO bulk phase. But, the presence of Li_2CO_3 on the surface of LLZTO is like sludge in the channels. Lithium ions can only enter and exit LLZTO particles from the thin layer of Li_2CO_3 and can only be transported in the LLZTO bulk phase (Fig. 8a), limiting the diffusion direction of lithium ions to the periphery, short of a crisscross lithium-ion conduction

network. Transforming Li_2CO_3 layer into active LiCoO_2 layer is like dredging the channels. Lithium ions can be freely transported not only in the bulk phase but also the surface of LLZTO@LCO , allowing the rapid lithium-ion transport paths to branch in any direction (Fig. 8b), which realizes uniform diffusion of lithium ions on the shallow and deep layer of the cathode. The rapid lithium-ion transport channel can be compared to an irrigation canal, in which the main channel is constructed along the high ionic conductivity area where more LLZTO@LCO particles are distributed, and the main channel branches to the surroundings to deliver lithium ions to various locations of the cathode, which greatly improves the transport efficiency of lithium ions. This transport mechanism allows lithium ions to be quickly and evenly distributed throughout the cathode, hence the LCO-LLZTO@LCO composite cathode exhibits the highest ionic conductivity compared with the other two cathodes.

Fig. 8 Lithium-ion transport paths of the $\text{LLZTO@Li}_2\text{CO}_3$ and LLZTO@LCO particles. TEM images of the **a** $\text{LLZTO@Li}_2\text{CO}_3$ and **b** LLZTO@LCO particles. (This figure is used as Fig. S12 in supplementary information)

The following revision has been made in our manuscript:

Page 12, line 20: Meanwhile, a thin buffer layer that is solid at room temperature was used to reduce the interface impedance between the electrolyte and the electrodes (Fig. S11).

Page 9, line 6: Characterization and ionic transport mechanism of the LCO-LLZTO@LCO composite cathode. To demonstrate the enhancement of LLZTO@LCO on the ionic transport of the composite cathode, the ionic conductivity of LCO-LLZTO@LCO and $\text{LCO-LLZTO@Li}_2\text{CO}_3$ composite cathodes were measured by EIS, respectively. Notably, the electronic conductor of Ketjen Black (KB) was not added to the cathode when measuring the ionic conductivity, and two pieces of carbon-containing aluminum foils were used as the blocking electrodes. The Al/Cathode without

KB/Al with a sandwich structure was measured in the frequency range of 1 MHz to 1 Hz. Fig. 4a shows the impedance plots of Al/Cathode without KB/Al. The cathodes exhibit an impedance behavior similar to the solid electrolyte, including an incomplete semicircle at a high frequency of 1 MHz and a slanted tail at low frequency,^{20, 21} which indicates that the cathodes without KB mainly have ionic conductivity rather than electronic conductivity. This is because the cathodes do not contain KB that dominates the electronic conduction, but the LiCoO₂ active material and the LLZTO ion conductor which dominate ionic conduction. The intercepts on the resistance axis are the ionic resistance (R_i) and the calculated results of the ionic conductivity are shown with blue histograms in Fig. 4c. The LCO-LLZTO@LCO composite cathode exhibits higher ionic conductivity of $5.5 \times 10^{-5} \text{ S cm}^{-1}$, improving the ionic conductivity of the LCO-LLZTO@Li₂CO₃ composite cathode by nearly 300%, which implies that the activated LLZTO@LCO promotes ionic transport inside cathode. This can be attributed to the decrease in the ionic transport resistance at the interfaces inside cathode, optimizing the ionic transport channels.

Additionally, the Al/Cathode with KB/Al was further used to investigate the mixed ionic and electronic conductivity of the LCO-LLZTO@LCO composite cathode. The semicircles exhibit no capacitive tail in the impedance plots of the Al/Cathode with KB/Al, as shown in Fig. 4b, indicating that the cathodes with KB are the mixed conductors, but the electronic resistance is much lower than the ionic resistance, and the intercepts on the resistance axis are the electronic resistance (R_e).^{22, 23} The results of the calculated electronic conductivity are shown with yellow histograms in Fig. 4c. It can be seen that the electronic conductivity of the LCO-LLZTO@LCO and LCO-LLZTO@Li₂CO₃ composite cathodes are almost the same, which can be ascribed to the fact that the KB plays a major role in electronic conduction inside cathodes. Combined with lower ionic conductivity and similar electronic conductivity of the pure LiCoO₂ cathode (Fig. S10), we can draw a conclusion that in the composite cathode, LLZTO particles can build rapid channels for lithium-ion transport, and the activated LLZTO@LCO surface further reduces the interface impedance between particles in the channels, enhancing ionic conduction inside cathode, but the electron conduction is almost not affected.

In order to explore the mechanism by which LLZTO@LCO particles enhance the transport of lithium ions inside cathode, the cross-sectional micromorphology of the cathodes was observed. As shown in Fig. 4d, before adding LLZTO to the LiCoO₂ cathode, the LiCoO₂ nanoparticles are

distributed on the cathode layer with a thickness of 15 μm , constructing a lithium-ion transport network. After LLZTO@Li₂CO₃ and LLZTO@LCO are introduced (Fig. 4e, f), they are distributed throughout the cathode and are in close contact with LiCoO₂ particles around, providing composite channels for lithium-ion transport. Significantly, LLZTO@Li₂CO₃ and LLZTO@LCO particles have large particle size and cross the cathode layer, which can construct rapid large-span channels for the transport of the lithium ions to the interior of the cathode. However, the Li₂CO₃ layer on the surface of LLZTO has low conductivity, which increases the interface impedance between the active material and the ion conductor particles. Conversely, LLZTO@LCO particles have an active surface in close contact with the active materials (Fig. S11), which is more conducive to the diffusion of lithium ions between LLZTO ionic conductor and LiCoO₂ active material.

Based on the above results, it can be concluded that the optimization of the transport channels for lithium ions by LLZTO@LCO is the reason for the improved ionic conductivity. A possible mechanism is provided in Fig. 4g. In the LiCoO₂ cathode without an ionic conductor, lithium ions diffuse into cathode through the ionic channels constructed by the active material LiCoO₂ with low ionic conductivity, which reduces the diffusion rate and diffusion depth of lithium ions, causing a large voltage polarization and limiting electrode reaction to occur in the shallow layer of the cathode. But, after LLZTO@Li₂CO₃ is added, large-span transport channels for lithium ions are formed around LLZTO@Li₂CO₃ particles, which can quickly transport lithium ions deeper into the LCO-LLZTO@Li₂CO₃ composite cathode. However, the presence of Li₂CO₃ on the surface of LLZTO is like sludge in the channels. Lithium ions can only enter and exit LLZTO particles from the thin layer of Li₂CO₃ and can only be transported to LiCoO₂ particles through the LLZTO bulk phase, limiting the diffusion direction of lithium ions to the periphery, short of a crisscross lithium-ion transport network. Transforming Li₂CO₃ layer into active LiCoO₂ layer is like dredging the channels. Lithium ions can be freely transported not only in the bulk phase but also the surface of LLZTO@LCO, allowing the rapid lithium-ion transport paths to branch in any direction due to the ion conductive of the entire LLZTO@LCO particle, which realizes uniform diffusion of lithium ions on the shallow and deep layer of the LCO-LLZTO@LCO composite cathode. The rapid lithium-ion transport channel can be compared to an irrigation canal, in which the main channel is constructed along the high ionic conductivity area where more LLZTO@LCO particles are distributed, and the main channel branches to the surroundings to deliver lithium ions to various

locations of the cathode, which greatly improves the transport efficiency of lithium ions. This transport mechanism allows lithium ions to be quickly and evenly distributed throughout the cathode, hence the LCO-LLZTO@LCO composite cathode exhibits higher ionic conductivity.

And the following description has been made in new supporting information:

Fig. S10: Fig. S10a shows the impedance plots of Al/Cathode without KB/Al. The intercepts on the resistance axis are the ionic resistance (R_i) of the cathodes. The ionic conductivity of the cathodes were calculated using $\sigma = \frac{1}{R} \frac{L}{A}$, where σ is the ionic conductivity of the cathode; R is the ionic resistance R_i ; L and A are the cathode thickness and measurement area, respectively. The calculated results of the ionic conductivity are shown with a blue line in Fig. S10b. The LCO-LLZTO@LCO composite cathode exhibits the highest ionic conductivity of $5.5 \times 10^{-5} \text{ S cm}^{-1}$. The LiCoO_2 cathode exhibits the lowest ionic conductivity of $0.8 \times 10^{-5} \text{ S cm}^{-1}$.

Additionally, the Al/Cathode with KB/Al was used to measure the electronic conductivity of the cathodes. It has been reported when both ionic and electron transport are significant to a mixed conductor, the impedance plot using the blocking electrodes is shown in Fig. S10c, and the equivalent circuit consists of an electronic current path and an ionic current path in parallel (inset in Fig. S10c). C_{geom} is the parallel plate capacitance of the two electrodes and C_{int} is the interface capacitance between electrodes and the intermediate medium. R_e is the electronic resistance, represented by the intercept (R_1) on the resistance axis at low frequencies. R_i is the ionic resistance, which in parallel with R_e is presented by the intercept (R_2) on the resistance axis at high frequencies.⁴ But, when the electronic conduction is dominant, the electron current will be much larger than the ionic current, resulting in a short circuit of the interface capacitance C_{int} , thus only presenting a semicircle in the impedance plot. Meanwhile, no capacitive tail appears at low frequency due to the shunt of the electronic current path.^{4,5} The conductivity of cathodes with KB were also calculated using $\sigma = \frac{1}{R} \frac{L}{A}$, where σ is the electronic conductivity of the cathode; R is the electronic resistance R_e , obtained from the intercept on the resistance axis as shown in Fig. S10d; L and A are the cathode thickness and measurement area, respectively. The results of the calculated electronic conductivity are shown with a yellow line in Fig. S10b. It can be seen that the electronic conductivity of the LCO, LCO-LLZTO@LCO and LCO-LLZTO@ Li_2CO_3 cathodes are almost the same ($\sim 1 \times 10^{-3} \text{ S cm}^{-1}$).

Fig. S13: As shown in Fig. S13a, the buffer layer is semi-fluid and has strong viscoelasticity at 80 °C, which limits the fluidity of the buffer layer, making it difficult to diffuse into the cathode. Meanwhile, it is solid at room temperature, and has high elasticity as well as can be processed into a film (Fig. S13b), which ensures the solid state structure of the assembled battery at room temperature. As shown in Fig. S13c, an ultra-thin buffer layer is formed on the cathode, and the boundary between the buffer layer and the cathode layer can be clearly observed, indicating that the buffer layer is not infiltrated into the cathode. Additionally, the energy dispersive X-ray (EDX) mapping analysis results that the N element of SCN from the buffer layer is only distributed on the buffer layer area above the cathode and is clearly layered with the Co element representing the cathode layer (Fig. S13d), exclude the possibility that the buffer layer can penetrate into the cathode. Therefore, in the battery in the manuscript, the ionic conductivity inside the composite cathode is only contributed by the LLZTO particles and active material LiCoO_2 . The buffer layer does not penetrate into the cathode.

3. As described in the experiment, the gel “was dripped on the electrode surface” , how much gel electrolyte was used? How deep it can penetrate? Comparison between LLZO and LCO-LLZO@LCO cathode with the gel electrolyte should be made.

Response: The keyword "dripped on" is incorrect, which should be corrected to "scraped on". As shown in Fig. 9a, the buffer layer with strong viscoelasticity is a semi-fluid at 80 °C, which makes it difficult to diffuse into the cathode and can only be coated on the surface of the cathode with a doctor blade. At the same time, it is solid at room temperature, and has high elasticity as well as can be processed into a film (Fig. 9b). Fig.9c shows the cross-sectional SEM image of the LCO-LLZTO@LCO composite cathode coated with buffer layer. The thickness of the buffer layer at the interface cannot be recognized due to its very thinness, and only the surface can be observed. Additionally, there is a clear boundary between the buffer layer and the cathode layer, which implies that the buffer layer has not been penetrated into the cathode. The energy dispersive X-ray (EDX) mapping analysis results that the N elements of SCN are only distributed on the buffer layer area on the surface of the cathode and are clearly layered with the Co elements of the cathode layer (Fig. 9d). Therefore, the above results demonstrate that the buffer layer has no effect on the internal structure of the cathode.

Fig. 9 Characterization of the buffer layer. a, b Optical photos of the buffer layer at 80 °C and room temperature. c Cross-sectional SEM image of the LCO-LLZTO@LCO composite cathode coated with buffer layer. d EDX analysis of the LCO-LLZTO@LCO composite cathode coated with buffer layer. (This figure is used as Fig. S13 in supplementary information)

The following revision has been made in our manuscript:

Page 12, line 20: Meanwhile, a thin buffer layer that is solid at room temperature was used to reduce the interface impedance between the electrolyte and the electrodes (Fig. S11).

And the following description has been made in new supporting information:

Fig. S13: As shown in Fig. S13a, the buffer layer is semi-fluid and has strong viscoelasticity at 80 °C, which limits the fluidity of the buffer layer, making it difficult to diffuse into the cathode. Meanwhile, it is solid at room temperature, and has high elasticity as well as can be processed into a film (Fig. S13b), which ensures the solid state structure of the assembled battery at room temperature. As shown in Fig. S13c, an ultra-thin buffer layer is formed on the cathode, and the boundary between the buffer layer and the cathode layer can be clearly observed, indicating that the buffer layer is not infiltrated into the cathode. Additionally, the energy dispersive X-ray (EDX) mapping analysis results that the N element of SCN from the buffer layer is only distributed on the buffer layer area above the cathode and is clearly layered with the Co element representing the cathode layer (Fig. S13d), exclude the possibility that the buffer layer can penetrate into the cathode. Therefore, in the battery in the manuscript, the ionic conductivity inside the composite cathode is only contributed by the LLZTO particles and active material LiCoO_2 . The buffer layer does not penetrate into the cathode.

4. An all-ceramic composite cathode by co-sintering is strongly suggested to demonstrate the advantage of LLZO@LCO particles.

Response: As reviewer's suggestion, an all-ceramic battery was tried to be prepared by the co-sintering method. Unfortunately, we found that low temperature sintering (below 600 °C to ensure no diffusion of elements) causes a huge interface impedance of 300 kΩ cm² (Fig. 10a), so that the battery cannot be cycled even at 100 °C. In view of the above results, we also tried to paste the cathode slurry onto the LLZTO piece directly, and dried it under a pressure. SnO₂ was used as the interface layer on the anode side. The interface impedance of the prepared battery was reduced to a certain extent, (Fig. 10b), and the battery can be charged for the first cycle at 100 °C, but was severely damaged immediately (Fig. 10c). This may be due to the volume change of the cathode after the battery is charged, resulting in cracking of cathode/electrolyte interface, thereby increasing the interface impedance again (Fig. 10b). It is disappointing that although we tried our best to attempt various methods, it is still unavailable to address the issue of large interfacial resistance in a limited time. In this case, a buffer layer between the solid electrolyte and the LCO-LLZTO@LCO cathode is still necessary. As mentioned above, we have ruled out the possibility that the buffer layer can affect the cathode performance through the EDX analysis. Therefore, the excellent performance of LCO-LLZTO@LCO, as the cathode itself, is unquestionable. We beg the reviewer to forgive us for failing to supplement the electrochemical performance of the all-ceramic battery.

Fig.10 Electrochemical performance of the all-ceramic batteries. **a** EIS of the all-ceramic battery prepared by co-sintering. **b, c** EIS and discharge/charge curves of the all-ceramic battery prepared by applying cathode slurry, respectively.

Author's Response to Reviewer #2

1. The use of the term peritectic and peritectoid is not correct. There is no gas byproduct in these reactions.

Response: Thank you very much for your suggestion. We also recognize the inaccuracy. Here, we correct “on-surface peritectoid reaction” to the “on-surface lithium donor reaction”. This concept is derived from the popular hydrogen transfer reactions, in which the hydrogen donor is used as hydrogen source. As shown in Fig. 11, Li_2CO_3 layer and LLZTO are defined as lithium donors to provide lithium sources for the surface transformation reaction of LLZTO particles. “On-surface lithium donor reaction” not only accurately describes the characteristics of the two transformation reactions during the synthesis of LLZTO@LCO, but also emphasizes the surface reaction of the LLZTO particle, which appropriately summarizes the two prominent features of the material synthesis in this article.

Fig.11 Schematic illustration of the on-surface lithium donor reaction to achieve interface homogeneity. (This figure is used as Fig. 2a in the manuscript)

The following revision has been made in our manuscript:

Title: On-surface lithium donor reaction enables decarbonated lithium garnets and compatible interfaces within cathodes

Page 6, line 21: Li_2CO_3 and LLZTO as the lithium donors together provide the lithium sources for the transformation reaction to form the complete LiCoO_2 coating on the surface of LLZTO.

Others: All the term “peritectoid reaction” have been corrected to the “lithium donor reaction”.

And the following description has been made in new supporting information:

Title: On-surface lithium donor reaction enables decarbonated lithium garnets and compatible

interfaces within cathodes

2. Indeed Li_2CO_3 is not good for the Li interface, but it literature did not necessarily imply its bad for forming cathode interfaces.

Response: Thank you very much for your comments. By supplementing experiments and citing references, we explain that the presence of Li_2CO_3 in cathode interfaces seriously deteriorate the cyclic stability of the solid-state battery.

As the reviewer stated, Li_2CO_3 cause a large impedance at the anode interface in the LLZO-based solid-state battery due to his poor wettability to lithium metal. Similarly, the Li_2CO_3 formed at the grain boundary also can decrease the overall ionic conductivity of the cathode, because Li_2CO_3 exhibits the ultralow ionic conductivity of $\sim 10^{-8} \text{ S cm}^{-1}$ even at 200°C ,⁶ which hinders the transport of lithium ions between the particles inside cathode. It is known that the high ionic conductivity of the cathode is beneficial to eliminate overcharge or overdischarge of the active materials in the local, and reduce the voltage polarization. In liquid batteries, the ionic transport channels of the cathode are mainly constructed by the liquid electrolyte, because a large amount of electrolyte penetrates into cathode. In this case, the presence of trace amounts of Li_2CO_3 in the cathode will not affect the ionic conductivity of the entire cathode, thus Li_2CO_3 with low ionic conductivity is sometimes still considered as a component for construction of the artificial solid electrolyte interface (SEI) layer to prevent the reaction between the cathode and the liquid electrolyte.⁷ But it has been reported that decomposition of Li_2CO_3 during working also can cause aggregation of discharge polarization and seriously deteriorate the cyclic stability of the battery.⁸⁻¹⁰ Compared with the liquid electrolyte, the solid-state electrolyte are completely non-wetting and the electrolyte cannot be immersed in the cathode, so the large ionic resistance inside cathode is a serious issue for the solid-state battery. In order to increase the ion conductivity of the cathode, the LLZTO ionic conductor is added to construct the lithium-ion transport channel. If there is a Li_2CO_3 layer with low ionic conductivity on the surface of LLZTO that plays a major role in conducting lithium ions, the ionic conductivity of the entire cathode will be greatly reduced, causing large voltage polarization and accelerating the destruction of the cathode structure.

In addition, by measuring the ionic conductivity of the cathodes, we are also confirmed the limitation of Li_2CO_3 on the transport of the lithium ions inside cathode. The ionic conductivity of LCO, LCO-LLZTO@LCO and LCO-LLZTO@ Li_2CO_3 cathodes without KB were measured by

electrochemical impedance spectroscopy (EIS) respectively. It should be noted that the electronic conductor Ketjen Black (KB) is not added in the measurement of the ionic conductivity. The cathode slurry without KB was coated on the carbon-containing aluminum foil (as the blocking electrode) with a diameter of 15.5 mm and another piece of aluminum foil was pressed on the surface of the cathode after semi-drying. Then the Al/Cathode without KB)/Al sandwich structure samples were dried under vacuum while pressed with a weight. The a.c. impedance was measured in the frequency range of 1 MHz to 1 Hz.

Fig. 12 Measurement of the ionic conductivity of the cathodes without KB. **a** EIS of the Al/Cathode without KB/Al. **b** The ionic conductivity of the cathodes.

Fig. 12a shows the impedance plots of Al/Cathode without KB/Al. The cathodes exhibit an impedance behavior similar to the solid electrolyte, including an incomplete semicircle at a high frequency of 1 MHz and a slanted tail at low frequency,^{2,3} which indicates that the KB-free cathodes mainly have ionic conductivity rather than electronic conductivity. This is because the cathodes do not contain KB that dominates the electronic conduction, but the LiCoO₂ active material and the LLZTO ion conductor which dominate ionic conduction. The intercepts on the resistance axis are the ionic resistance (R_i) of the cathodes. The ionic conductivity of the cathodes were calculated using $\sigma = \frac{1}{R} \frac{L}{A}$, where σ is the ionic conductivity of the cathode; R is the ionic resistance R_i ; L and A are the cathode thickness and measurement area, respectively. The calculated results of the ionic conductivity are shown in Fig. 12b. The LCO-LLZTO@LCO composite cathode exhibits the highest ionic conductivity of $5.52 \times 10^{-5} \text{ S cm}^{-1}$, which is higher than that of the LCO-LLZTO@Li₂CO₃ composite cathode ($1.74 \times 10^{-5} \text{ S cm}^{-1}$), which implying that Li₂CO₃ at the interfaces inside cathode reduces the ionic conductivity of the cathode, but the activated LLZTO surface obtained by converting Li₂CO₃ to LiCoO₂ can decrease the lithium-ion transport resistance

at the interface. Significantly, the ionic conductivity of both composite cathodes is higher than the pure LiCoO_2 cathode ($0.8 \times 10^{-5} \text{ S cm}^{-1}$).

Fig. 13 DEMS analysis of LLZTO@ Li_2CO_3 in composite cathodes. (This figure is provided by Fig. 6a in the manuscript)

Besides, it has reported that the Li_2CO_3 impurities inside cathode will be decomposed under high voltage.¹¹ This phenomenon was also observed in our differential electrochemical mass spectrometry analysis (DEMS) test. Fig. 13 shows charge curve (top) and corresponding CO_2 emission (bottom) for the LCO-LLZTO@ Li_2CO_3 composite cathode. Notably, the intensity of CO_2 began to increase when charged to about 4.0 V (vs Li/Li^+), which is considered to be due to the decomposition of Li_2CO_3 in LLZTO@ Li_2CO_3 , consistent with previous observations that Li_2CO_3 was decomposed above 4.0 V.^{11, 12} The decomposition of Li_2CO_3 in the cathode during working can reduce the coulombic efficiency and damage the structure of the cathode, deteriorating the cyclic stability of the battery.

Therefore, Li_2CO_3 not only is not good for the Li interface, but also seriously damages the cathode interfaces of the solid-state battery due to its ionic insulation and high-voltage decomposition.

The following revision has been made in our manuscript:

Page 9, line 6: Characterization and ionic transport mechanism of the LCO-LLZTO@LCO composite cathode. To demonstrate the enhancement of LLZTO@LCO on the ionic transport of the composite cathode, the ionic conductivity of LCO-LLZTO@LCO and LCO-LLZTO@ Li_2CO_3 composite cathodes were measured by EIS respectively. Notably, the electronic conductor Ketjen Black (KB) is not added to the cathode in the measurement, and carbon-containing aluminum foil is used as the blocking electrode. The Al/Cathode without KB/Al with a sandwich structure was

measured in the frequency range of 1 MHz to 1 Hz. Fig. 4a shows the impedance plots of Al/Cathode without KB/Al. The cathodes exhibit an impedance behavior similar to the solid electrolyte, including an incomplete semicircle at a high frequency of 1 MHz and a slanted tail at low frequency,^{20, 21} which indicates that the cathodes without KB mainly have ionic conductivity rather than electronic conductivity. This is because the cathodes do not contain KB that dominates the electronic conduction, but the LiCoO₂ active material and the LLZTO ion conductor which dominate ionic conduction. The intercepts on the resistance axis are the ionic resistance (R_i) and the calculated results of the ionic conductivity are shown with blue histograms in Fig. 4c. The LCO-LLZTO@LCO composite cathode exhibits higher ionic conductivity of 5.5×10⁻⁵ S cm⁻¹, improving the ionic conductivity of the LCO-LLZTO@Li₂CO₃ composite cathode by nearly 300%, which implies that the activated LLZTO@LCO promotes ionic transport inside cathode. This can be attributed to the decrease in the ionic transport resistance at the interface between particles inside cathode, optimizing the ionic transport channels.

And the following description has been made in new supporting information:

Fig. S10: Fig. S10a shows the impedance plots of Al/Cathode without KB/Al. The intercepts on the resistance axis are the ionic resistance (R_i) of the cathodes. The ionic conductivity of the cathodes were calculated using $\sigma = \frac{1}{R} \frac{L}{A}$, where σ is the ionic conductivity of the cathode; R is the ionic resistance R_i; L and A are the cathode thickness and measurement area, respectively. The calculated results of the ionic conductivity are shown with a blue line in Fig. S10b. The LCO-LLZTO@LCO composite cathode exhibits the highest ionic conductivity of 5.5×10⁻⁵ S cm⁻¹. The LiCoO₂ cathode exhibits the lowest ionic conductivity of 0.8×10⁻⁵ S cm⁻¹.

3. *The biggest concern is the scientific process and presentation of results:*

a. What is the science here? Is it how the layer of Li_2CO_3 enables a high performance interface between LCO and LLZTO? If so, there is no SEM data on the microstructure showing what the electrodes look like. There is not electrochemical data that measures the interface resistance.

Response: Thanks for your suggestion. In order to explore the effect mechanism of the LLZTO surface layer on the electrochemical performance, the ionic and electronic conductivity and SEM data of the cathodes were measured.

The electronic and ionic conductivity of LCO, LCO-LLZTO@LCO and LCO-LLZTO@ Li_2CO_3 cathodes with or without KB were measured by electrochemical impedance spectroscopy (EIS) respectively. It should be noted that the electronic conductor Ketjen Black (KB) is not added in the measurement of the ionic conductivity. The cathode slurry with or without KB was coated on the carbon-containing aluminum foil (as the blocking electrode) with a diameter of 15.5 mm and another piece of aluminum foil was pressed on the surface of the cathode after semi-drying. Then the Al/Cathode with or without KB/Al sandwich structure samples were dried under vacuum while pressed with a weight. The a.c. impedance was measured in the frequency range of 1 MHz to 1 Hz.

Fig. 14a shows the impedance plots of Al/Cathode without KB/Al. The cathodes exhibit an impedance behavior similar to the solid electrolyte, including an incomplete semicircle at a high frequency of 1 MHz and a slanted tail at low frequency,^{2,3} which indicates that the cathodes without KB mainly have ionic conductivity rather than electronic conductivity. This is because the cathodes do not contain KB that dominates the electronic conduction, but the LiCoO_2 active material and the LLZTO ion conductor which dominate ionic conduction. The intercepts on the resistance axis are the ionic resistance (R_i) and the calculated results of the ionic conductivity are shown with blue line in Fig. 14c. The LCO-LLZTO@LCO composite cathode exhibits higher ionic conductivity of $5.5 \times 10^{-5} \text{ S cm}^{-1}$, improving the ionic conductivity of the pure LiCoO_2 cathode by nearly 650%, which implies that the LLZTO@LCO ionic conductor accelerates the transport of lithium ions to the other side of the cathode. Simultaneously, the activated LLZTO surface obtained by converting Li_2CO_3 to LiCoO_2 reduces the ionic transport resistance at the interface between the LiCoO_2 and LLZTO particles inside cathode, as a result, the ionic conductivity of the LCO-LLZTO@LCO composite cathode is further improved compared with that of the LCO-LLZTO@ Li_2CO_3 composite

cathode ($1.74 \times 10^{-5} \text{ S cm}^{-1}$).

Fig. 14 Characterization and ionic transport mechanism of the cathodes. **a** EIS of the Al/Cathode without KB/Al. **b** EIS of the Al/Cathode with KB/Al. **c** The ionic and electronic conductivity of the cathodes. **d-f** Cross-sectional SEM images of the LiCoO_2 , $\text{LCO-LLZTO@Li}_2\text{CO}_3$ and LCO-LLZTO@LCO cathodes, separately. **g** Schematic illustration of the ionic transport mechanism inside cathodes. (Parts of this figure are used as Fig. 4 in the manuscript)

Additionally, the Al/Cathode with KB/Al also was used to investigate the mixed ionic and electronic conductivity of the cathodes. The semicircles without capacitive tail in the impedance plots of the Al/Cathode with KB/Al, as shown in Fig. 14b, explain that the cathodes with KB are the mixed conductors but the electronic resistance is much lower than the ionic resistance, and the intercepts on the resistance axis are the electronic resistance (R_e).^{4, 5} The results of the calculated electronic conductivity are shown with yellow line in Fig. 14c. It can be seen that the electronic conductivity of the LiCoO_2 , LCO-LLZTO@LCO and $\text{LCO-LLZTO@Li}_2\text{CO}_3$ composite cathodes are almost the same ($\sim 1 \times 10^{-3} \text{ S cm}^{-1}$), which can be ascribed to the fact that KB plays a major role in electronic conduction inside cathodes. Combined with the EIS results of the cathodes without KB,

we can draw a conclusion that in the composite cathode, LLZTO particles can build rapid channels for lithium-ion transport, and the activated LLZTO@LCO surface further reduces the interface impedance between particles in the channels, enhancing ionic conduction inside cathode, but the electron conduction is almost not affected.

In order to explore the mechanism by which LLZTO@LCO particles enhance the transport of lithium ions inside cathode, the cross-sectional micromorphology of the cathodes were observed. As shown in Fig. 14d, before adding LLZTO to the LiCoO₂ cathode, the LiCoO₂ nanoparticles are distributed on the cathode layer with a thickness of 15 μm, constructing a lithium-ion transport network. When LLZTO@Li₂CO₃ is added, the LLZTO@Li₂CO₃ particles are distributed throughout the cathode and are in close contact with LiCoO₂ nanoparticles around (Fig. 14e), providing composite channels for lithium-ion transport. Significantly, as can be seen from Fig. 14e and 14f, after LLZTO@Li₂CO₃ or LLZTO@LCO are introduced to the cathodes separately, they with a large particle size cross the cathode layer, which can construct rapid large-span channels for the transport of the lithium ions to the interior of the cathode. However, the Li₂CO₃ layer on the surface of LLZTO with low conductivity will increase the interface impedance between the active material and the ionic conductor particles. Conversely, LLZTO@LCO particles have an active surface in close contact with the active materials, which is more conducive to the diffusion of lithium ions between LLZTO ionic conductor and LiCoO₂ active material.

Based on the above results, it can be concluded that the optimization of the transport channels for lithium ions by LLZTO@LCO may be the reason for the improved ionic conductivity, thus a possible mechanism is provided in Fig. 14g. In the pure LiCoO₂ cathode without an ionic conductor, lithium ions diffuse into the inside of cathode through the ionic channels constructed by the active material LiCoO₂ with low ionic conductivity, which can reduce the diffusion rate and diffusion depth of lithium ions, causing a large voltage polarization and limiting electrode reaction to occur in the shallow layer of the cathode. But, after LLZTO@Li₂CO₃ is added, large-span transport channels for lithium ions are formed around LLZTO@Li₂CO₃ particles, which can quickly transport lithium ions deeper into the cathode. But, the presence of Li₂CO₃ on the surface of LLZTO is like sludge in the channels. Lithium ions can only enter and exit LLZTO particles from the thin layer of Li₂CO₃ and can only be transported in the LLZTO bulk phase (Fig. 15a), limiting the diffusion direction of lithium ions to the periphery, short of a crisscross lithium-ion conduction network. Transforming

Li_2CO_3 layer into active LiCoO_2 layer is like dredging the channels. Lithium ions can be freely transported not only in the bulk phase but also the surface of LLZTO@LCO , allowing the rapid lithium-ion transport paths to branch in any direction (Fig. 15b), which realizes uniform diffusion of lithium ions on the shallow and deep layer of the cathode. The rapid lithium-ion transport channel can be compared to an irrigation canal, in which the main channel is constructed along the high ionic conductivity area where more LLZTO@LCO particles are distributed, and the main channel branches to the surroundings to deliver lithium ions to various locations of the cathode, which greatly improves the transport efficiency of lithium ions. This transport mechanism allows lithium ions to be quickly and evenly distributed throughout the cathode, so the LCO-LLZTO@LCO composite cathode exhibits the highest ionic conductivity compared to the other two cathodes.

Fig. 15 Lithium-ion transport paths of the $\text{LLZTO@Li}_2\text{CO}_3$ and LLZTO@LCO particles
 TEM images of the **a** $\text{LLZTO@Li}_2\text{CO}_3$ and **b** LLZTO@LCO particles. (This figure is used as Fig. S12 in supplementary information)

The following revision has been made in our manuscript:

Page 9, line 6: Characterization and ionic transport mechanism of the LCO-LLZTO@LCO composite cathode. To demonstrate the enhancement of LLZTO@LCO on the ionic transport of the composite cathode, the ionic conductivity of LCO-LLZTO@LCO and $\text{LCO-LLZTO@Li}_2\text{CO}_3$ composite cathodes were measured by EIS respectively. Notably, the electronic conductor Ketjen Black (KB) was not added to the cathode in the measurement, and carbon-containing aluminum foil was used as the blocking electrode. The Al/Cathode without KB/Al with a sandwich structure was measured in the frequency range of 1 MHz to 1 Hz. Fig. 4a shows the impedance plots of Al/Cathode without KB/Al. The cathodes exhibit an impedance behavior similar to the solid electrolyte, including an incomplete semicircle at a high frequency of 1 MHz and a slanted tail at low

frequency,^{20, 21} which indicates that the cathodes without KB mainly have ionic conductivity rather than electronic conductivity. This is because the cathodes do not contain KB that dominates the electronic conduction, but the LiCoO₂ active material and the LLZTO ion conductor which dominate ionic conduction. The intercepts on the resistance axis are the ionic resistance (R_i) and the calculated results of the ionic conductivity are shown with blue histograms in Fig. 4c. The LCO-LLZTO@LCO composite cathode exhibits higher ionic conductivity of $5.5 \times 10^{-5} \text{ S cm}^{-1}$, improving the ionic conductivity of the LCO-LLZTO@Li₂CO₃ composite cathode by nearly 300%, which implies that the activated LLZTO@LCO promotes ionic transport inside cathode. This can be attributed to the decrease in the ionic transport resistance at the interfaces inside cathode, optimizing the ionic transport channels. Additionally, the Al/Cathode with KB/Al also was used to investigate the mixed ionic and electronic conductivity of the LCO-LLZTO@LCO composite cathode. The semicircles without capacitive tail in the impedance plots of the Al/Cathode with KB/Al, as shown in Fig. 4b, explain that the cathodes with KB are the mixed conductors but the electronic resistance is much lower than the ionic resistance, and the intercepts on the resistance axis are the electronic resistance (R_e).^{22, 23} The results of the calculated electronic conductivity are shown with yellow histograms in Fig. 4c. It can be seen that the electronic conductivity of the LCO-LLZTO@LCO and LCO-LLZTO@Li₂CO₃ composite cathodes are almost the same, which can be ascribed to the fact that KB plays a major role in electronic conduction inside cathodes. Combined with lower ionic conductivity and similar electronic conductivity of the pure LiCoO₂ cathode (Fig. S10), we can draw a conclusion that in the composite cathode, LLZTO particles can build rapid channels for lithium-ion transport, and the activated LLZTO@LCO surface further reduces the interface impedance between particles in the channels, enhancing ionic conduction inside cathode, but the electron conduction is almost not affected.

In order to explore the mechanism by which LLZTO@LCO particles enhance the transport of lithium ions inside cathode, the cross-sectional micromorphology of the cathodes were observed. As shown in Fig. 4d, before adding LLZTO to the LiCoO₂ cathode, the LiCoO₂ nanoparticles are distributed on the cathode layer with a thickness of 15 μm , constructing a lithium-ion transport network. After LLZTO@Li₂CO₃ and LLZTO@LCO are introduced (Fig. 4e, f), they are distributed throughout the cathode and are in close contact with LiCoO₂ particles around, providing composite channels for lithium-ion transport. Significantly, LLZTO@Li₂CO₃ and LLZTO@LCO particles

with a large particle size cross the cathode layer, which can construct rapid large-span channels for the transport of the lithium ions to the interior of the cathode. However, the Li_2CO_3 layer on the surface of LLZTO with low conductivity will increase the interface impedance between the active material and the ion conductor particles. Conversely, LLZTO@LCO particles have an active surface in close contact with the active materials (Fig. S11), which is more conducive to the diffusion of lithium ions between LLZTO ionic conductor and LiCoO_2 active material.

Based on the above results, it can be concluded that the optimization of the transport channels for lithium ions by LLZTO@LCO may be the reason for the improved ionic conductivity, thus a possible mechanism is provided in Fig. 4g. In the LiCoO_2 cathode without an ionic conductor, lithium ions diffuse into cathode through the ionic channels constructed by the active material LiCoO_2 with low ionic conductivity, which can reduce the diffusion rate and diffusion depth of lithium ions, causing a large voltage polarization and limiting electrode reaction to occur in the shallow layer of the cathode. But, after LLZTO@ Li_2CO_3 is added, large-span transport channels for lithium ions are formed around LLZTO@ Li_2CO_3 particles, which can quickly transport lithium ions deeper into the LCO-LLZTO@ Li_2CO_3 composite cathode. But, the presence of Li_2CO_3 on the surface of LLZTO is like sludge in the channels. Lithium ions can only enter and exit LLZTO particles from the thin layer of Li_2CO_3 and can only be transported to LiCoO_2 particles through the LLZTO bulk phase, limiting the diffusion direction of lithium ions to the periphery, short of a crisscross lithium-ion transport network. Transforming Li_2CO_3 layer into active LiCoO_2 layer is like dredging the channels. Lithium ions can be freely transported not only in the bulk phase but also the surface of LLZTO@LCO, allowing the rapid lithium-ion transport paths to branch in any direction due to the ion conductive of the entire LLZTO@LCO particle, which realizes uniform diffusion of lithium ions on the shallow and deep layer of the LCO-LLZTO@LCO composite cathode. The rapid lithium-ion transport channel can be compared to an irrigation canal, in which the main channel is constructed along the high ionic conductivity area where more LLZTO@LCO particles are distributed, and the main channel branches to the surroundings to deliver lithium ions to various locations of the cathode, which greatly improves the transport efficiency of lithium ions. This transport mechanism allows lithium ions to be quickly and evenly distributed throughout the cathode, so the LCO-LLZTO@LCO composite cathode exhibits higher ionic conductivity.

And the following description has been made in new supporting information:

Fig. S10: Fig. S10a shows the impedance plots of Al/Cathode without KB/Al. The intercepts on the resistance axis are the ionic resistance (R_i) of the cathodes. The ionic conductivity of the cathodes were calculated using $\sigma = \frac{1}{R_i} \frac{L}{A}$, where σ is the ionic conductivity of the cathode; R_i is the ionic resistance; L and A are the cathode thickness and measurement area, respectively. The calculated results of the ionic conductivity are shown with a blue line in Fig. S10b. The LCO-LLZTO@LCO composite cathode exhibits the highest ionic conductivity of $5.5 \times 10^{-5} \text{ S cm}^{-1}$. The LiCoO_2 cathode exhibits the lowest ionic conductivity of $0.8 \times 10^{-5} \text{ S cm}^{-1}$.

Additionally, the Al/Cathode with KB/Al was used to measure the electronic conductivity of the cathodes. It has been reported when both ionic and electron transport are significant to a mixed conductor, the impedance plot using the blocking electrodes is shown in Fig. S10c, and the equivalent circuit consists of an electronic current path and an ionic current path in parallel (inset in Fig. S10c). C_{geom} is the parallel plate capacitance of the two electrodes and C_{int} is the interface capacitance between electrodes and the intermediate medium. R_e is the electronic resistance, represented by the intercept (R_1) on the resistance axis at low frequencies. R_i is the ionic resistance, which in parallel with R_e is presented by the intercept (R_2) on the resistance axis at high frequencies. But, when the electronic conduction is dominant, the electron current will be much larger than the ionic current, resulting in a short circuit of the interface capacitance C_{int} , thus only presenting a semicircle in the impedance plot. Meanwhile, no capacitive tail appears at low frequency due to the shunt of the electronic current path.^{4,5} The conductivity of cathodes with KB were also calculated using $\sigma = \frac{1}{R_e} \frac{L}{A}$, where σ is the electronic conductivity of the cathode; R_e is the electronic resistance, obtained from the intercept on the resistance axis as shown in Fig. S10d; L and A are the cathode thickness and measurement area, respectively. The results of the calculated electronic conductivity are shown with a yellow line in Fig. S10b. It can be seen that the electronic conductivity of the LCO, LCO-LLZTO@LCO and LCO-LLZTO@ Li_2CO_3 cathodes are almost the same ($\sim 1 \times 10^{-3} \text{ S cm}^{-1}$).

b. The most concerning is the lack of clarity on whether or not a liquid electrolyte is used in the electrochemical data? Figure 4 (Fig. 3 in the revised manuscript) uses liquid electrolyte to characterize the behavior of the cathode. Figure 5 uses LiTFSI and succinonitrile. Why the change in electrolytes?

c. Why was a liquid electrolyte used at all? The way the introduction was written, the impression is given that the goal is to eliminate the use of liquid electrolyte in the cathode by reducing interface resistance.

Response: Thank you very much for your comments. It is so sorry that we did not specify the necessity of using liquid electrolyte in Fig. 3 and of using succinonitrile interface layer in Fig. 5 in the manuscript. So we make supplementary explanations as follows, and make supplements in the manuscript. In addition, in view of the similarity of the above two questions, we answer them together.

Fig. 3c and 3d in the manuscript are to investigate lithium-intercalated activity of the LiCoO₂ layer in LLZTO@LCO. LLZTO@LCO is used as the active material to prepare cathode with binder and conductive carbon, without addition of any other active materials. Because there are fewer active materials in the cathode, the use of liquid electrolyte is necessary, and it is beneficial to depict the redox peaks of the synthesized LiCoO₂ layer in more details, so that it can be accurately compared with the characteristic redox peaks and charging/discharging voltage platforms (~3.9 V) of commercial LiCoO₂. If a solid-state electrolyte is used, the voltage polarization caused by the larger interface impedance will shift the redox potentials of LiCoO₂, deviating from the characteristic charging/discharging voltage platforms of commercial LiCoO₂, which will cause misunderstanding of the lithium-intercalated behavior of the synthesized LiCoO₂ layer. At the same time, large voltage polarization will broaden the redox peaks and cover part of the redox signal (Fig. 16), making an incomplete analysis of LLZTO@LCO. Therefore, we used a liquid battery to investigate lithium-intercalated activity of the LLZTO@LCO. Differently, Fig. 5 in the manuscript is to prove that LLZTO@LCO particles can address the issue of the low ionic conductivity inside cathode in the solid state battery. But the large interface resistance between the solid electrolyte and the electrodes makes the solid-state battery difficult to work, so we introduced a succinonitrile interface layer in the electrolyte/electrode interface to reduce the interface resistance, making the solid-state battery can operate normally to verify the superiority of the LCO-LLZTO@LCO

composite cathode. Notably, succinonitrile-based layer is not used as an electrolyte but as a solid interface layer to reduce the interface impedance of the solid-state battery.

Fig. 16 CVs of the batteries with a different electrolyte. The cathode contains only LLZTO@LCO, conductive carbon and binder.

The following revision has been made in our manuscript:

Page 8, line 19: It should be noted that the liquid electrolyte is used here to characterize the lithium-intercalated behavior of the LLZTO@LCO in more details, so that it can be accurately compared with the characteristic redox peaks and charging/discharging voltage platforms (~3.9 V) of commercial LiCoO₂.

d. Where is the interface resistance in the EIS in Figure 5? I would think the main reason for doing EIS is to measure the interface resistance to validate that the approach works.

Response: Thank you very much for your reminder. As the reviewer mentioned, EIS is used to measure the interface resistance of the batteries to confirm the lower interface resistance in the battery employing an LCO-LLZTO@LCO cathode. Fig. 17 shows the impedance plots of the batteries using an LCO-LLZTO@Li₂CO₃ cathode and an LCO-LLZTO@LCO cathode, respectively, after 100 cycles. It can be seen that the impedance plot includes an incomplete curve at a high frequency, a semicircular at middle frequency and a tail at low frequency. The intercept on the real axis at high frequencies is the bulk resistance of the LLZTO piece (R_b), and the intercept on the real axis at middle frequencies is the total resistance of the R_b and the interface resistance (R_i). R_i including the electrolyte/electrodes interface resistance and the interface resistance inside cathode, but the resistance of the two interfaces are not easily separated and calculated. Apparently, R_b in the two batteries is almost the same ($250 \Omega \text{ cm}^2$), but the battery with an LCO-LLZTO@LCO cathode exhibits a smaller R_i of $850 \Omega \text{ cm}^2$ compared with the battery with an LCO-LLZTO@Li₂CO₃ cathode ($1300 \Omega \text{ cm}^2$), which can be attributed to the lower interface resistance inside the LCO-LLZTO@LCO cathode, because the same interface layer was used results in the same interface resistance between the electrolyte and the electrodes.

Fig. 17 EIS of the solid-state batteries after 100 cycles. (This figure is provided by Fig. 5e, f in the manuscript)

The following revision has been made in our manuscript:

Page 13, line 14: which can be attributed to the lower interfacial resistance inside the LCO-

LLZTO@LCO cathode due to the presence of optimized ion-conducting channels.

Author's Response to Reviewer #3

1. According to Fig. 2b, the lithium carbonate layer and cobalt oxide undergo the on-surface peritectoid reaction to produce lithium cobalt oxide. At the same time, the lithium source inside LLZTO also reacts slightly with the cobalt oxide to generate the lithium cobalt oxide layer. In these two reactions, which one dominates the lithium cobalt oxide layer?

Response: Thank you very much for your comments. The mutual contact of the reactants is the premise of the solid-phase reaction. In the mixed raw materials of Co_3O_4 and LLZTO that coated with Li_2CO_3 , the direct contact of Co_3O_4 and Li_2CO_3 allows the chemical reactions to begin, so the conversion reaction of Co_3O_4 and Li_2CO_3 to LiCoO_2 occurs preferentially. Meanwhile, Fig. 18 shows the sintering results of LLZTO and Co_3O_4 in different ratios. It can be seen that when a small amount of Co_3O_4 is added to the reaction, there is no Li_2CO_3 in the sintered product, indicating that the reaction between Li_2CO_3 and Co_3O_4 has proceeded. When the amount of Co_3O_4 continues to increase, the $\text{La}_2\text{Zr}_2\text{O}_7$ phase appears in the sintered product, which is generated by the reaction of the lithium source of LLZTO with Co_3O_4 . The above results indicate that in the lithium donor reaction, the reaction of Co_3O_4 and Li_2CO_3 occurs first, and the reaction of the lithium source with Co_3O_4 can be controlled to proceed by adjusting the proportion of Co_3O_4 .

Fig. 18 XRD patterns of LLZTO@Li₂CO₃ and Co₃O₄ sintered at different ratios.

2. Figure 5 shows the DEMS results of carbon dioxide emission. The cells were assembled by lithium cobalt oxide active materials, mixed with LLZTO electrolytes with and without lithium cobalt oxide layer, separately. How about that of the commercial lithium cobalt oxide active materials without the LLZTO electrolyte as the ionic conductor?

Response: Based on the reviewer's suggestion, we supplemented the DEMS data of the LiCoO_2 cathode. Fig. 19a shows charge curve (top) and corresponding CO_2 emission (bottom) for the commercial LiCoO_2 cathode. It can be seen that, similar to the LCO-LLZTO@LCO composite cathode (Fig. 18c), CO_2 is not released in the LiCoO_2 cathode. This is due to the absence of Li_2CO_3 in the two cathodes, which also emphasizes the superiority of the LLZTO@LCO material, that is, it can improve the ionic conductivity of the cathode without introducing impurities.

Fig. 19 DEMS analysis of the cathodes. a-c Charge curves (top) and corresponding CO_2 emission (bottom) of the pure LCO, LCO-LLZTO@ Li_2CO_3 and LCO-LLZTO@LCO cathode, separately.

3. The buffer layer does not use a liquid solvent or a drying process. What is the state of the buffer layer in the solid-state lithium battery? Detailed information should be provided.

Response: As shown in Fig. 20a, the buffer layer with strong viscoelasticity is a semi-fluid at 80 °C, which makes it difficult to diffuse into the cathode and can only be coated on the surface of the cathode with a doctor blade. At the same time, it is solid at room temperature, and has high elasticity as well as can be processed into a film (Fig. 20b), which ensures the solid state structure of the assembled battery at room temperature. Fig. 20c shows the cross-sectional SEM image of the LCO-LLZTO@LCO composite cathode coated with buffer layer. The thickness of the buffer layer at the interface cannot be recognized due to its very thinness, and only the surface can be observed. The energy dispersive X-ray (EDX) mapping analysis results that the N elements of SCN are only distributed on the buffer layer area on the surface of the cathode and are clearly layered with the Co elements of the cathode layer (Fig. 20d), confirm that the buffer layer does not penetrate into the cathode and has no effect on the internal structure of the cathode.

Fig. 20 Characterization of the buffer layer. a, b Optical photos of the buffer layer at 80 °C and room temperature. c Cross-sectional SEM image of the LCO-LLZTO@LCO composite cathode coated with buffer layer. d EDX analysis of the LCO-LLZTO@LCO composite cathode coated with buffer layer. (This figure is used as Fig. S13 in supplementary information)

The following revision has been made in our manuscript:

Page 12, line 20: Meanwhile, a thin buffer layer that is solid at room temperature was used to reduce

the interface impedance between the electrolyte and the electrodes (Fig. S11).

And the following description has been made in new supporting information:

Fig. S13: As shown in Fig. S13a, the buffer layer is semi-fluid and has strong viscoelasticity at 80 °C, which limits the fluidity of the buffer layer, making it difficult to diffuse into the cathode. Meanwhile, it is solid at room temperature, and has high elasticity as well as can be processed into a film (Fig. S13b), which ensures the solid state structure of the assembled battery at room temperature. As shown in Fig. S13c, an ultra-thin buffer layer is formed on the cathode, and the boundary between the buffer layer and the cathode layer can be clearly observed, indicating that the buffer layer is not infiltrated into the cathode. Additionally, the energy dispersive X-ray (EDX) mapping analysis results that the N element of SCN from the buffer layer is only distributed on the buffer layer area above the cathode and is clearly layered with the Co element representing the cathode layer (Fig. S13d), exclude the possibility that the buffer layer can penetrate into the cathode. Therefore, in the battery in the manuscript, the ionic conductivity inside the composite cathode is only contributed by the LLZTO particles and active material LiCoO_2 . The buffer layer does not penetrate into the cathode.

4. Typos. Page 2, line 18: *maintains*; Page 2, line 21: *the intimate*; Page 3, line 35, *in the recent year*; Page 6, line 103: *after the first sintering*.

Response: Thank you very much for reminding us these mistakes, we have corrected them in the manuscript.

References

- 1 Dai, Y., Lu, P., Cao, Z., Campbell, C. T. & Xia, Y. The physical chemistry and materials science behind sinter-resistant catalysts. *Chem. Soc. Rev.* **47**, 4314-4331 (2018).
- 2 Ma, Q. *et al.* Viscoelastic and nonflammable interface design-enabled dendrite-free and safe solid lithium metal batteries. *Adv. Energy Mater.* **9**, 1803854 (2019).
- 3 Huang, X. *et al.* Influence of $\text{La}_2\text{Zr}_2\text{O}_7$ additive on densification and Li^+ conductivity for Ta-doped $\text{Li}_7\text{La}_3\text{Zr}_2\text{O}_{12}$ garnet. *J. Minerals, Metals & Mater. Soc.* **68**, 2593-2600 (2016).
- 4 Wang, C., & Hong, J. Ionic/electronic conducting characteristics of LiFePO_4 cathode materials the determining factors for high rate performance. *Electrochem. Solid State Lett.* **10**, 65-69 (2007).
- 5 Huggins, R. Simple method to determine electronic and ionic components of the conductivity in mixed conductors a review. *Ionics* **8**, 300-313 (2002).
- 6 Dissanayake, M., & Mellander B. E. Phase diagram and electrical conductivity of the Li_2SO_4 - Li_2CO_3 system. *Solid State Ionics* **21**, 279-285 (1986).
- 7 Dai, X. *et al.* Extending the high-voltage capacity of LiCoO_2 cathode by direct coating of the composite electrode with Li_2CO_3 via magnetron sputtering. *J. Phys. Chem. C* **120**, 422-430 (2015).
- 8 Bi, Y. *et al.* Stability of Li_2CO_3 in cathode of lithium ion battery and its influence on electrochemical performance. *Rsc Adv.* **6**, 19233-19237 (2016).
- 9 Liu, W. *et al.* Nickel-rich layered lithium transition-metal oxide for high-energy lithium-ion batteries. *Angew. Chem. Int. Ed.* **54**, 4440-4457 (2015).
- 10 Zhuang, G. *et al.* Li_2CO_3 in $\text{LiNi}_{0.8}\text{Co}_{0.15}\text{Al}_{0.05}\text{O}_2$ cathodes and its effects on capacity and power. *J. Power Sources* **134**, 293-297 (2004).
- 11 Meini, S. *et al.* Recharge ability of Li-air cathodes pre-filled with discharge products using an ether-based electrolyte solution: implications for cycle-life of Li-air cells. *Phys. Chem. Chem. Phys.* **27**, 11478-11493 (2013).
- 12 Jung, R., Strobl, P., Maglia, F., Stinner, C., & Gasteiger, H. A. Temperature dependence of oxygen release from $\text{LiNi}_{0.6}\text{Mn}_{0.2}\text{Co}_{0.2}\text{O}_2$ (NMC622) cathode materials for Li-ion batteries. *J. Electrochem. Soc.* **165**, A2869-A2879 (2018).

REVIEWER COMMENTS

Reviewer #1 (Remarks to the Author):

This paper reports a method to remove the Li_2CO_3 on the LLZTO surface by forming a LiCoO_2 layer, which might help improve the preparation of composite cathode in garnet-based solid-state batteries. However, the following concerns remain to be solved:

1. In paragraph 1 on page 7, the authors attributed the LiCoO_2 formation partially to the reaction between the Co_3O_4 and the evaporated Li from LLZTO by an experiment (Fig. S6). However, it is difficult to tell from Fig. S6 that LLZTO decomposition is part of the reason for LiCoO_2 formation because a LLZTO@ Li_2CO_3 sample was used. A Li_2CO_3 -free sample should be used to eliminate the contribution of Li_2CO_3 .

2. In paragraph 2 on page 8, the authors claim that no Li_2CO_3 phase was formed for LLZTO@LCO after exposing to air for four months. This is confusing because Li_2CO_3 has been widely known to be able to form on positive active material including LiCoO_2 in air [K. Matsumoto, et al. Journal of Power Sources 81-82 (1999) 558-561; T. Hayashi et al. Journal of Power Sources 354 (2017) 41-47].

Reviewer #2 (Remarks to the Author):

Q1: A peritectoid reaction is the reaction two distinct solids that cool to form a third distinct solid. What is being to a peritectoid reaction is still incorrect. Ions are transferring. It's not that phases are forming. My suggestion. Don't call it a phase transformation. Moreover, what was simply suggested related to the use of the term peritectoid. What the authors responded with is not relevant to this question.

Q2: The response to the effect of lithium carbonate on the surface of the LCO is not adequate. First of all, Al is non-blocking, therefore interface and/or charge transfer reactions will or could be obscured by the reaction between Li and Al. Second, what would be expected if there were lithium carbonate affecting the interface resistance. There should be a control where they demonstrate what happens if lithium carbonate does increase interface resistance. In addition, there should be replicates or triplicates to show this is reproducible.

Q3: I do not understand why the authors integrate the use of conductive additive into the response to the question. The question is about the ionic conductivity between LCO and LLZTO. Adding/subtracting ketjen black will likely obscure any efforts to isolate the ionic conductivity between the LCO and LLZTO.

The SEM if Figure 14 d-f are good for macroscopic analysis, but I think the more significant aspect is the quality and nature of the bond between LCO and LLZTO. That interface can not be observed at that length scale.

Q4: Regarding the use of a liquid or waxy gel electrolyte like succinonitrile still does not make sense. If the point of the lithium carbonate on LCO/LLZTO is to reduce the interface resistance, it seems like a situation where the issue with one interface is to use another interface.

Q5: The EIS analysis to measure the interface resistances still is not adequate. The frequencies and capacitances must be used in the analysis to confirm what is being measured is the interface resistance.

Reviewer #3 (Remarks to the Author):

Authors have addressed all of my comments. It can be accepted as is.

Author's Response to Reviewer #1:

1. In paragraph 1 on page 7, the authors attributed the LiCoO_2 formation partially to the reaction between the Co_3O_4 and the evaporated Li from LLZTO by an experiment (Fig. S6). However, it is difficult to tell from Fig. S6 that LLZTO decomposition is part of the reason for LiCoO_2 formation because an $\text{LLZTO@Li}_2\text{CO}_3$ sample was used. A Li_2CO_3 -free sample should be used to eliminate the contribution of Li_2CO_3 .

Response: Thank you very much for your comments. As your suggestion, Li_2CO_3 -free LLZTO was used to prove that the evaporated Li from LLZTO is part of the reason for LiCoO_2 formation. Initially, the freshly prepared LLZTO powder was analyzed by FTIR, we found that there is almost no Li_2CO_3 on the surface of LLZTO (Fig. 1a). To avoid the generation of Li_2CO_3 when LLZTO is exposed to air, LLZTO and Co_3O_4 were mixed and ground in an argon-filled glove box and sintered in an argon atmosphere. Fig. 1b shows the XRD pattern of the sintered products of Li_2CO_3 -free LLZTO and Co_3O_4 . It can be seen that $\text{La}_2\text{Zr}_2\text{O}_7$ is formed after sintering, indicating the presence of lithium volatilization in LLZTO. At the same time, LiCoO_2 is also generated, confirming that Co_3O_4 reacts with evaporated Li to generate LiCoO_2 .

Fig. 1 Sintering results of Li_2CO_3 -free LLZTO and Co_3O_4 . **a** FTIR spectra of LLZTO prepared freshly or stored in air for a long time. **b** XRD pattern of Li_2CO_3 -free LLZTO and Co_3O_4 sintered at 600°C for 4h. (This figure is used as Fig. S6 in supplementary information)

The following revision has been made in our manuscript:

Page 7, line 2: Second, we designed an experiment of sintering Li_2CO_3 -free LLZTO and Co_3O_4 under the same condition. The $\text{La}_2\text{Zr}_2\text{O}_7$ and LiCoO_2 were still found in the sintered products,

indicating that lithium volatilization exists in LLZTO during the sintering process, and the volatilized lithium can react with Co_3O_4 to form LiCoO_2 (Fig. S6).

And the following description has been made in new supporting information:

Fig. S6: Initially, the freshly prepared LLZTO powder was analyzed by FTIR, we found that there is almost no Li_2CO_3 on the surface of LLZTO (Fig. S6a). To avoid the generation of Li_2CO_3 when LLZTO is exposed to air, LLZTO and Co_3O_4 were mixed and ground in an argon-filled glove box and sintered in an argon atmosphere. Fig. S6b shows the XRD pattern of the sintered products of Li_2CO_3 -free LLZTO and Co_3O_4 . It can be seen that $\text{La}_2\text{Zr}_2\text{O}_7$ is formed after sintering, indicating the presence of lithium volatilization in LLZTO. At the same time, LiCoO_2 is also generated, which confirms that Co_3O_4 reacts with evaporated Li and generates LiCoO_2 .

2. In paragraph 2 on page 8, the authors claim that no Li_2CO_3 phase was formed for LLZTO@LCO after exposing to air for four months. This is confusing because Li_2CO_3 has been widely known to be able to form on positive active material including LiCoO_2 in air [K. Matsumoto, et al. *Journal of Power Sources* 81-82 (1999) 558-561; T. Hayashi et al. *Journal of Power Sources* 354 (2017) 41-47].

Response: We quite agree with the reviewer's opinion that it is necessary to consider whether Li_2CO_3 is formed on the surface of LiCoO_2 . We carefully read the two pieces of literature listed by the reviewer. We performed a supplementary experiment and concluded that LiCoO_2 is stable in air. Here we explain the stability of the LiCoO_2 against the air as follows:

The first literature listed by the reviewer confirms that $\text{LiNi}_{0.81}\text{Co}_{0.16}\text{Al}_{0.03}\text{O}_2$ can react with atmospheric CO_2 and generate Li_2CO_3 even at room temperature. The conversion quantities of this reaction are calculated from the quantified CO_2 values in the article. But the author mentioned in a comparative experiment that LiCoO_2 was also subjected to the atmospheric exposure test, but the detection of CO_2 was within measurement error, which indicates that atmospheric CO_2 is almost not consumed, in other words, LiCoO_2 does not react with CO_2 to generate Li_2CO_3 . This result implies that LiCoO_2 is relatively stable in the air compared with $\text{LiNi}_{0.81}\text{Co}_{0.16}\text{Al}_{0.03}\text{O}_2$, corresponding to our experimental result that the Li_2CO_3 phase was not formed on the LiCoO_2 layer after exposing LLZTO@LCO to air for four months. The reason is also mentioned in their introduction that the basicity of LiNiO_2 -based active materials is higher than that of LiCoO_2 active material and easily cause its degradation by atmospheric CO_2 .

The second literature listed by the reviewer describes that the amorphous lithium tungsten oxide layer on the surface of the LiCoO_2 can effectively prevent the side reaction between LiCoO_2 and the electrolyte, thereby eliminating the formation of Li_2CO_3 on the surface of the LiCoO_2 . It is worth noting that the Li_2CO_3 layer on the surface of LiCoO_2 is generated by the reaction of LiCoO_2 and the liquid electrolyte during charging and discharging, rather than during atmospheric exposure. Therefore, their test results do not conflict with our conclusion.

Additionally, Fig. 2 shows the FTIR spectrum of LiCoO_2 (purchased from Aladdin) stored in air for about two years. It can be seen that there are no peaks of Li_2CO_3 at 863 cm^{-1} and 1438 cm^{-1} , indicating that Li_2CO_3 is not formed on the surface of LiCoO_2 , which further confirms that LiCoO_2

is stable in air. The peaks in the wavenumber range of 400-480 cm^{-1} are related to the occupation of lithium ions in the tetrahedron of LiCoO_2 .¹

Fig. 2 FTIR spectrum of LiCoO_2 after long-term storage in the atmosphere. The inset shows an enlarged image in the wavenumber range of 400-480 cm^{-1} .

Author's Response to Reviewer 2:

1. A peritectoid reaction is the reaction two distinct solids that cool to form a third distinct solid. What is being to a peritectoid reaction is still incorrect. Ions are transferring. It's not that phases are forming. My suggestion. Don't call it a phase transformation. Moreover, what was simply suggested related to the use of the term peritectoid. What the authors responded with is not relevant to this question.

Response: Thank you very much for your comments. We quite agree with the reviewer's opinion that it is not correct to describe the conversion process of Li_2CO_3 by citing the peritectoid reaction.

Initially, we only considered that the change process of one solid phase surrounding another solid phase in the term peritectoid reaction is similar to our transformation process, in which the solid phase of Li_2CO_3 is transformed on the surface of the LLZTO solid phase. The LiCoO_2 layer derives from the Li_2CO_3 and Co_3O_4 phase, and this transformation eliminates the complex third phase inside cathodes. However, we ignored the production of gas and didn't accurately grasp that the peritectoid reaction is essentially a phase transformation process.

In fact, the peritectoid reaction is a physical phase transformation that is cooling of a binary alloy in which one solid phase surrounds the other solid phase to create a completely different and single solid phase. Differently, the conversion process of Li_2CO_3 does not involve the physical phase transformation, but a chemical reaction. Therefore, as the reviewer's suggestion, we don't call it a phase transformation, but a chemical reaction defined as a lithium donor reaction. Meanwhile, all descriptions related to the peritectoid reaction and phase transformation have been revised to the lithium donor reaction in the manuscript and supporting information.

2. The response to the effect of lithium carbonate on the surface of the LCO is not adequate. First of all, Al is non-blocking, therefore interface and/or charge transfer reactions will or could be obscured by the reaction between Li and Al. Second, what would be expected if there were lithium carbonate affecting the interface resistance. There should be a control where they demonstrate what happens if lithium carbonate does increase interface resistance. In addition, there should be replicates or triplicates to show this is reproducible.

Response: Thanks for your suggestion. Firstly, the reviewer mentioned Li_2CO_3 on the surface of LCO, which may mean Li_2CO_3 on the surface of LLZTO, because there is no Li_2CO_3 on the surface of LCO in this article. Secondly, as stated by the reviewer, Al foil is a non-blocking electrode, which may obscure the interface and/or charge transfer reactions due to the alloying of Li and Al. At this moment, gold foil (the inset in Fig. 3) instead of Al foil was used as a blocking electrode to re-measure the ionic conductivity of composite cathodes without ketjen black. Fig. 3 shows the re-measured ionic conductivity of the LCO-LLZTO@LCO, LCO-LLZTO@ Li_2CO_3 , and LiCoO_2 cathodes, and the ionic conductivity of each cathode was tested for three times. The results are similar to that of Al as an electrode. The average ionic conductivity of the LCO-LLZTO@LCO cathode is the highest, and that of the LiCoO_2 cathode is the lowest. The measurement error may mainly come from the different thickness of the prepared cathode films.

Fig. 3 Ionic conductivity of the LCO-LLZTO@LCO, LCO-LLZTO@ Li_2CO_3 , and LiCoO_2 cathodes without Ketjen black (KB). The inset shows the gold foil used as a blocking electrode. (This figure is used as Fig. S10 in supplementary information)

Secondly, to comparatively analyze what happens when Li_2CO_3 increases the interface resistance inside cathode, freshly prepared LLZTO powder without Li_2CO_3 was also used as an ion conductor

to prepare composite cathode. And the effect of Li_2CO_3 on the lithium-ion transfer kinetics of the cathodes was analyzed by testing lithium-ion apparent diffusion coefficient. Fig. 4a-c show the cyclic voltammetry (CV) profiles of the LLZTO (without Li_2CO_3)-containing LiCoO_2 (LCO-LLZTO), LCO-LLZTO@ Li_2CO_3 and LCO-LLZTO@LCO cathodes at different scan rates. The lithium-ion apparent diffusion coefficient can be calculated according to the Randles-Sevcik equation:²

$$I_p = 2.68 \times 10^5 n^{3/2} A C D^{1/2} \nu^{1/2} \quad (1)$$

Where I_p is the peak current (A); n is the charge-transfer number of the redox reaction. A is the area of the cathode plate (cm^2); C is the lithium-ion concentration in LiCoO_2 cathode ($0.051 \text{ mol cm}^{-3}$); D is the lithium-ion diffusion coefficient ($\text{cm}^2 \text{ s}^{-1}$); ν is the scan rate (V s^{-1}). I_p is linearly related to $\nu^{1/2}$ and the value of $I_p/\nu^{1/2}$ can be obtained from the linear fitting results as 0.01663, 0.01391 and 0.01744 for LCO-LLZTO, LCO-LLZTO@ Li_2CO_3 and LCO-LLZTO@LCO cathodes (Fig. 4d-f). The corresponding lithium-ion apparent diffusion coefficient could be calculated to be $1.85 \times 10^{-13} \text{ cm}^2 \text{ s}^{-1}$ for LCO-LLZTO cathode, $1.28 \times 10^{-13} \text{ cm}^2 \text{ s}^{-1}$ for LCO-LLZTO@ Li_2CO_3 cathode and $2.04 \times 10^{-13} \text{ cm}^2 \text{ s}^{-1}$ for LCO-LLZTO@LCO cathode. Compared with LCO-LLZTO@ Li_2CO_3 cathode, the lithium-ion diffusion coefficient of the LCO-LLZTO cathode is increased by about 45%, which indicates that the production of Li_2CO_3 on the surface of LLZTO will slow down the diffusion of lithium ions inside cathode. This is because Li_2CO_3 with high ionic resistance increases the interface resistance between particles inside cathode. Additionally, after removing Li_2CO_3 by converting it into LiCoO_2 , the lithium-ion diffusion coefficient is increased by about 59%. This may be due to the formation of a low resistance interface between LiCoO_2 and LLZTO inside cathode.

Fig. 4 Lithium-ion apparent diffusion coefficient of the cathodes. CV profiles of **a** LCO-LLZTO, **b** LCO-LLZTO@Li₂CO₃, and **c** LCO-LLZTO@LCO cathodes at different scan rates. Peak current as a function of the square root of the scan rate of **d** LCO-LLZTO, **e** LCO-LLZTO@Li₂CO₃, and **f** LCO-LLZTO@LCO cathodes. (Fig. 5b, c, e, and f are used as Fig. 4a-c in the manuscript.)

The following revision has been made in our manuscript:

Page 9, line 5: In order to investigate the effect of LLZTO@LCO on the lithium-ion transfer kinetics of the cathode, the lithium-ion apparent diffusion coefficient was tested by performing cyclic voltammetry (CV) measurements. Fig. 4a, b show the CV profiles of the LCO-LLZTO@LCO and LCO-LLZTO@Li₂CO₃ cathodes at different scan rates. The lithium-ion apparent diffusion coefficient can be calculated according to the Randles-Sevcik equation²⁰:

$$I_p = 2.68 \times 10^5 n^{3/2} A C D^{1/2} v^{1/2} \quad (1)$$

Where I_p is the peak current (A); n is the charge-transfer number of the redox reaction; A is the area of the cathode (cm²); C is the lithium-ion concentration in LiCoO₂ cathode (0.051 mol cm⁻³); D is the lithium-ion diffusion coefficient (cm² s⁻¹); v is the scan rate (V s⁻¹). I_p is linearly related to $v^{1/2}$ and the value of $I_p/v^{1/2}$ can be obtained from the linear fitting results as 0.01744 and 0.01391 for LCO-LLZTO@LCO and LCO-LLZTO@Li₂CO₃ cathodes (Fig. 4c). The lithium-ion apparent diffusion coefficient could be calculated to be 2.04×10^{-13} cm² s⁻¹ for LCO-LLZTO@LCO cathode and 1.28×10^{-13} cm² s⁻¹ for LCO-LLZTO@Li₂CO₃ cathode. Significantly, after Li₂CO₃ is converted to LiCoO₂, the lithium-ion diffusion coefficient of the cathode is increased by about 59%. This is because the activated LLZTO@LCO promotes ionic transport inside cathode, decreasing the ionic

transport resistance inside cathode. (Fig. S10).

And the following description has been made in new supporting information:

Fig. S10: The cathode slurry without KB was coated on the gold foil (as the blocking electrode) with a diameter of 15.5 mm, and another piece of gold foil was pressed on the surface of the cathode after semi-drying. Then the Au/Cathode without KB/Au sandwich structure samples were dried at 80 °C for 8h, and measured by electrochemical impedance spectroscopy in the frequency range of 1 MHz to 1 Hz. The ionic conductivity of the cathodes were calculated using $\sigma = \frac{1}{R} \frac{L}{A}$, where σ is the ionic conductivity of the cathode; R is the ionic resistance obtained from the intercept on the resistance axis in the EIS; L and A are the cathode thickness and measurement area, respectively. The average ionic conductivity of the LCO-LLZTO@LCO cathode is the highest, and that of the LiCoO₂ cathode is the lowest. The measurement error may mainly come from the different thickness of the prepared cathode films.

3. I do not understand why the authors integrate the use of conductive additive into the response to the question. The question is about the ionic conductivity between LCO and LLZTO. Adding/subtracting ketjen black will likely obscure any efforts to isolate the ionic conductivity between the LCO and LLZTO.

The SEM if Figure 14 d-f are good for macroscopic analysis, but I think the more significant aspect is the quality and nature of the bond between LCO and LLZTO. That interface can not be observed at that length scale.

Response: We are very sorry for showing redundant experiment results in the question. In the last response, testing the conductivity of the cathode with KB or without KB separately was to clarify that the conversion of Li_2CO_3 to LiCoO_2 can increase the ionic conductivity of the cathode without affecting its electronic conductivity. But, as stated by the reviewer, the question is about the ionic conductivity between LiCoO_2 and LLZTO and measuring the conductivity of the KB-added cathode is nonsense, because the ion conductivity between LiCoO_2 and LLZTO will be masked by the electronic conductivity caused by KB. In view of this, the conductivity measurement of the cathode with KB is deleted from the manuscript, and only the ion-conductivity measurement of the cathode without KB is retained.

In addition, as mentioned by the reviewer, the quality and nature of the bond between LiCoO_2 and LLZTO may also significantly affect the ionic conductivity of the cathode. Therefore, to observe the interface between LiCoO_2 and LLZTO, an LLZTO@LCO particle underwent TEM lamella preparation by using a focused ion beam (FIB) system. Fig. 5a shows the preparation process of the lamella sample. Firstly, platinum (Pt) and carbon (C) protection layers are deposited onto the surface of an LLZTO@LCO particle by FIB system before lamella preparation to avoid damage from the Ga-ion milling. Then, the block is milled around the LLZTO@LCO particle by FIB to take out the preliminary sample (cuboid area drawn by red line). Subsequently, the sample is gradually milled along the milling surfaces on both sides of the sample to thin it. After the milling surfaces are polished, the LLZTO@LCO lamella with a thickness of about 100 nm is prepared. Fig. 5b shows the TEM image of the LLZTO@LCO lamella. It can be seen that there are many LiCoO_2 nano-fragments distributed along the surface of LLZTO. This distribution is also clearly observed in the bright-field (BF) scanning transmission electron microscopy (STEM) image (Fig. 5d). But, it also can be seen that a small part of the LLZTO surface is not covered by LiCoO_2 , This may be caused

by the damage of Ga-ion beam to the surface layer during lamella preparation. Additionally, from the HRTEM image (Fig. 5c) corresponding to the enclosed area in Fig. 5b, we can observe that there is a relatively chaotic atomic arrangement at the interface between LLZTO and LiCoO₂ within a range of about 3 nm, implying the atoms are interlaced at the junction of LLZTO and LiCoO₂, which leads to a tightly connected interface and facilitates the transfer of lithium ions between LLZTO and LiCoO₂, thereby increasing the diffusion rate of lithium ions inside cathode. The lattice spacing of 0.528 nm agrees with the (112) facets of LLZTO, and the lattice spacing of 0.245 and 0.201 nm agrees with the (101) and (104) facets of LiCoO₂ respectively.

Fig. 5 Characterization of the interface between LLZTO and LiCoO₂ in LLZO@LCO particle.

a Schematic illustration of the LLZTO@LCO lamella preparation by using FIB system. **b** TEM, HRTEM and BF-STEM images of the LLZTO@LCO lamella. (This figure is used as Fig. S12 in supplementary information)

The following revision has been made in our manuscript:

Page 10, line 12: Conversely, LLZTO@LCO particles not only have an active surface in close contact with the active materials (Fig. S12), but also a tight interface at the junction of its own LLZTO and LiCoO₂ (Fig. S13), which is more conducive to the diffusion of lithium ions inside cathode.

And the following description has been made in new supporting information:

Fig. S12: Fig. S12a shows the preparation process of LLZTO@LCO lamella sample. Firstly, platinum (Pt) and carbon (C) protection layers are deposited onto the surface of an LLZTO@LCO particle by FIB system before lamella preparation to avoid damage from the Ga-ion milling. Then,

the block is milled around the LLZTO@LCO particle by FIB to take out the preliminary sample (cuboid area drawn by red line). Subsequently, the sample is gradually milled along the milling surfaces on both sides of the sample to thin it. After the milling surfaces are polished, the LLZTO@LCO lamella with a thickness of about 100 nm is prepared. Fig. S12b shows the TEM image of the LLZTO@LCO lamella. It can be seen that there are many LiCoO₂ nano-fragments distributed along the surface of LLZTO. This distribution is also clearly observed in the bright-field (BF) scanning transmission electron microscopy (STEM) image (Fig. S12d). But, it also can be seen that a small part of the LLZTO surface is not covered by LiCoO₂, This may be caused by the damage of Ga-ion beam to the surface layer during lamella preparation. Additionally, in the HRTEM image (Fig. S12c) corresponding to the enclosed area in Fig. 5b, the lattice spacing of 0.528 nm agrees with the (112) facets of LLZTO, and the lattice spacing of 0.245 and 0.201 nm agrees with the (101) and (104) facets of LiCoO₂ respectively. Importantly, we can observe that there is a relatively chaotic atomic arrangement at the interface between LLZTO and LiCoO₂ within a range of about 3 nm, implying the atoms at the junction of LLZTO and LiCoO₂ are interlaced, which leads to a tightly connected interface and facilitates the transfer of lithium ions between LLZTO and LiCoO₂, thereby increasing the diffusion rate of lithium ions inside cathode.

4. Regarding the use of a liquid or waxy gel electrolyte like succinonitrile still does not make sense. If the point of the lithium carbonate on LCO/LLZTO is to reduce the interface resistance, it seems like a situation where the issue with one interface is to use another interface.

Response: Thank you very much for your comments. It is so sorry that we did not clarify the necessity of the succinonitrile (SCN) interlayer in the solid-state battery. In view of this, here we explain the purpose of the SCN interlayer and emphasize it in the manuscript.

In this article, the conversion of Li_2CO_3 on the surface of LLZTO particle to LiCoO_2 addresses the issue of the micro-interface that is the interface between the LiCoO_2 active materials and LLZTO ionic conductor particles inside cathode (as shown in the left side of Fig. 6). Differently, SCN electrolyte film is used as an interlayer to optimize the macro-interface that is the interface between the LLZTO electrolyte piece and the electrode plates (as shown on the right side of Fig. 6). SCN interlayer is necessary for the solid cell because the large resistance of the macro-interface caused by the inherent solid-to-solid contact will make the cell unable to work even at 100 °C if it is not used (Fig.7). In this case, the use of SCN electrolyte film is to enable the solid cell to operate normally, so as to verify the superiority of the LCO-LLZTO@LCO cathode.

Fig. 6 Major interfaces in the solid-state battery. The micro-interface is the interface between particles inside cathode. The macro-interface is the interface between the electrolyte piece and the electrode plates.

Fig.7 Electrochemical performance of the solid-state battery without SCN interlayer. a Discharge/charge curves and **b** EIS of the solid-state battery without SCN interlayer. The solid cell without SCN interlayer cannot operate because the large interface resistance derived from the macro-interface.

Fig. 8 Limitation of SCN interlayer in the macro-interface. a, b Optical photos of the SCN interlayer at 80 °C and room temperature. **c** Cross-sectional SEM image of the LCO-LLZTO@LCO composite cathode coated with SCN interlayer. **d** EDX analysis of the LCO-LLZTO@LCO composite cathode coated with SCN interlayer. (This figure is shown in Fig. S14 in supplementary information)

Importantly, the regulation of these two interfaces does not interfere with each other in this article. As shown in Fig. 8a, the SCN interlayer with strong viscoelasticity is a semi-fluid at 80 °C, which makes it difficult to diffuse into the cathode. At the same time, it is solid at room temperature and has high elasticity as well as can be processed into a film (Fig. 8b), which is conducive to limit it to the macro-interface, so as not to affect the micro-interface. Fig.8c shows the cross-sectional SEM image of the LCO-LLZTO@LCO composite cathode coated with the SCN interlayer. It can be seen

that there is a clear boundary between the SCN interlayer and cathode layer, implying that the SCN has not been penetrated into the cathode. The energy dispersive X-ray (EDX) mapping analysis results further confirm this fact. The N elements of SCN are only distributed on the interlayer area and are clearly layered with the Co elements of the cathode layer (Fig. 8d). Therefore, the above results demonstrate that the SCN interlayer in the macro-interface does not interfere with the micro-interface inside cathode.

The following revision has been made in our manuscript:

Page 12, line 2: Meanwhile, a thin buffer layer that is solid at room temperature was used to reduce the interface impedance between the electrolyte piece and the electrode plates (Fig. S14).

5. The EIS analysis to measure the interface resistances still is not adequate. The frequencies and capacitances must be used in the analysis to confirm what is being measured is the interface resistance.

Response: Thanks for your suggestion. Here, the frequencies and capacitances were used in the EIS analysis. Fig. 9 shows EIS of the SSBs with an LCO-LLZTO@LCO and LCO-LLZTO@Li₂CO₃ cathode, separately. It can be seen that the impedance plot includes an incomplete semicircle in the high-frequency region, a semicircle in the middle frequency region, and a tail in low frequency region. The corresponding equivalent circuit is shown in the inset of Fig. 9. R_b is for bulk resistance of LLZTO electrolyte piece. R_g is for grain boundary impedance of LLZTO electrolyte piece. R_{int} and CPE_{int} are for overall interface resistance and double-layer capacitance inside the battery. The Warburg impedance (W) and CPE_d are for diffusion impedance.³ The simulated values are shown in Table 1. In the middle frequency region (794 Hz), the total interfacial R_{int} in the battery with an LCO-LLZTO@LCO cathode is 600 Ω cm² after 180 cycles, lower than 988 Ω cm² in the battery with an LCO-LLZTO@Li₂CO₃ cathode. The corresponding CPE_{int} capacitance is 3.8×10⁻⁶ F cm² and 5.0×10⁻⁶ F cm² respectively. The smaller interface resistance is mainly due to the optimized ion transfer channels of the LLZTO@LCO particles.

Fig. 9 Equivalent circuit analysis of EIS for the SSBs with an LCO-LLZTO@LCO and LCO-LLZTO@Li₂CO₃ cathode, separately. (This figure is used as Fig. S17 in supplementary information)

Table 1 Analysis results of equivalent circuits for EIS

Composite cathode		R_b $\Omega \text{ cm}^2$	R_g $\Omega \text{ cm}^2$	CPE_g $\times 10^{-9}$ $F \text{ cm}^2$	R_{int} $\Omega \text{ cm}^2$	CPE_{int} $\times 10^{-6}$ $F \text{ cm}^2$	CPE_d $\times 10^{-3}$ $F \text{ cm}^2$
LCO- LLZTO@LCO	Initial	24	166	1.7	360	5.2	7.6
	After 180 cycles	35	202	6.2	600	3.8	9.3
LCO- LLZTO@Li ₂ CO ₃	Initial	23	120	2.7	435	3.5	6.6
	After 180 cycles	30	340	1.7	988	5.0	6.5

The following revision has been made in our manuscript:

Page 12, line 16: Fig. 5e and 5f show the electrochemical impedance spectroscopy (EIS) of the batteries. It can be seen that the impedance plot includes an incomplete semicircle in the high-frequency region, a semicircle in the middle-frequency region, and a tail in low-frequency region, in which the semicircle in the middle frequency region corresponds to the overall interface resistance (R_{int}) inside the battery. The interface R_{int} in the battery with an LCO-LLZTO@LCO cathode is $600 \Omega \text{ cm}^2$ after 180 cycles, lower than $988 \Omega \text{ cm}^2$ in the battery with an LCO-LLZTO@Li₂CO₃ cathode (Fig. S17). The smaller interface resistance is mainly due to the optimized ion transfer channels of the LLZTO@LCO particles.

And the following description has been made in new supporting information:

Fig. S17: Fig. S17 shows EIS of SSBs before cycling and after 180 cycles. The corresponding equivalent circuit is shown in the inset. R_b is for bulk resistance of the LLZTO electrolyte piece. R_g is for grain boundary impedance of the LLZTO electrolyte piece. R_{int} and CPE_{int} are for overall interface resistance and double-layer capacitance inside the battery. The Warburg impedance (W) and CPE_d are for diffusion impedance. The simulated values are shown in Table 1.

Table 1 Analysis results of equivalent circuits for EIS.

Composite cathode		R_b $\Omega \text{ cm}^2$	R_g $\Omega \text{ cm}^2$	CPE_g $\times 10^{-9}$ $F \text{ cm}^2$	R_{int} $\Omega \text{ cm}^2$	CPE_{int} $\times 10^{-6}$ $F \text{ cm}^2$	CPE_d $\times 10^{-3}$ $F \text{ cm}^2$
LCO- LLZTO@LCO	Initial	24	166	1.7	360	5.2	7.6
	After 180 cycles	35	202	6.2	600	3.8	9.3
LCO- LLZTO@Li ₂ CO ₃	Initial	23	120	2.7	435	3.5	6.6
	After 180 cycles	30	340	1.7	988	5.0	6.5

References:

- 1 Rao, K. *et al.* Infrared spectroscopic study of LiCoO₂ thin films. *J. Solid State Chem.* **165**, 42-47 (2002).
- 2 Wang, L. P. *et al.* Ameliorating the interfacial problems of cathode and solid-state electrolytes by interface modification of functional polymers. *Adv. Energy Mater.* **8**, 1801528.1-1801528.8 (2018).
- 3 Liu, B. *et al.* Garnet solid electrolyte protected Li-metal batteries. *ACS Appl. Mater. Interfaces* **9**, 18809-18815 (2017).

REVIEWER COMMENTS

Reviewer #1 (Remarks to the Author):

Authors have addressed all of my comments. It can be accepted as it is.

Reviewer #2 (Remarks to the Author):

Re: Reviewer 2, Comment 1: Regarding Reviewer #2 Question #2, It is good to see that the authors used gold as the electrode material, though it's not purely blocking, but ok if the potential stays above the alloying potential. However, the ac impedance data requires rigorous analysis to extrapolate ionic conductivity. Interpreting mixed conduction is particularly challenging as electronic conductivity can superimpose on ionic conductivity. The authors need to show the ac impedance data, how it was modeled with an equivalent circuit, and show all three measurements for each measurement. In addition, 1 MHz was the maximum frequency used, thus it is not likely that all the salient transport phenomena are captured here and this has a significant impact on the interpretation and outcomes.

Re: Reviewer 2, Comment 2: The comment was adequately addressed.

Re: Reviewer 2, Comment 3: The authors are to be commended for the TEM analysis as it is complex and laborious. They have satisfied the question regarding the quality and nature of the bond between LCO and LLZTO. Well done. However, the other part of this response was practical and more important in terms of the impact of this study. The quality of the interface regarding the resistance remains unanswered. If the authors haven't read it already, the work by Goodenough on LCO-LLZO indicated that Co diffuses into LLZO thereby compromising the interface kinetics. Thus, while the authors did a great analysis with the TEM, the question about the interface kinetics remains. Can the authors use micro-electrodes along with their nicely FIB'ed samples to achieve this?

Re: Reviewer 2, Comment 4: The use of SCN is now clear. Thank you for clarifying.

Re: Reviewer 2, Comment 5: Regarding the EIS of the full cells: 1) is it correct to assume Li metal was one electrode? If so, where does the Li interface resistance show up in the EIS analysis. 2) why do the grain boundary and interface resistances change for both cells? 3) the @lithium carbonate cell EIS data seems to change in the EIS data between the bulk and grain boundary. Why is this so? 4) Can the authors justify why they used this equivalent circuit or who they will cite for this equivalent circuit? The bulk resistance requires a capacitive or constant-phase-element otherwise there would be no capacitive signal or imaginary impedance values.

Author's Response to Reviewer #1:

Comment: *Authors have addressed all of my comments. It can be accepted as it is.*

Response: We are grateful to this review for recommending acceptance.

Author's Response to Reviewer #2:

Response: We thank the review for endorsing the suitability of our work for publication in Nature Communications. The following is a point-by-point response to the reviewers' comments.

Comment 1: *It is good to see that the authors used gold as the electrode material, though it's not purely blocking, but ok if the potential stays above the alloying potential. However, the ac impedance data requires rigorous analysis to extrapolate ionic conductivity. Interpreting mixed conduction is particularly challenging as electronic conductivity can superimpose on ionic conductivity. The authors need to show the ac impedance data, how it was modeled with an equivalent circuit, and show all three measurements for each measurement. In addition, 1 MHz was the maximum frequency used, thus it is not likely that all the salient transport phenomena are captured here and this has a significant impact on the interpretation and outcomes.*

Response: These comments do make sense. First of all, as the reviewer stated, gold can be considered as a blocking electrode here, because only a bias voltage of 10 mV was applied in the impedance test so that the cells underwent a small voltage polarization, which is far from the lithium-gold alloy potential of 0.3 V.¹ Next, interpreting mixed conduction is subject to debate. To make it clear, the conductivity we measured by EIS includes two contributions: ionic and electronic, neither of which is separated from the other. However, the observation of the tail in the case of blocking Au electrodes (Fig. 1a) demonstrates that there is no significant electronic conductivity in the KB-free cathode.^{2,3} In other words, ionic conduction dominates the transport. The same phenomenon exists in the previously reported conductivity measurement of the solid electrolyte by EIS.^{4,5,6}

Fig. 1 EIS analysis of cathodes without KB in the frequency range of 1 MHz~0.1 Hz. EISs of Au/Cathode without KB/Au samples. **b** Conductivity of the cathodes without KB. (This figure is used as Fig. S10 in Supplementary Information)

Fig. 1a shows the impedance plots of Au/Cathode without KB/Au samples, including LiCoO_2 , $\text{LCO-LLZTO@Li}_2\text{CO}_3$ and LCO-LLZTO@LCO cathodes. Each impedance plot shows an incomplete semicircle at high frequency and a tail at low frequency, which is similar to the typical EIS of blocking-electrode/solid electrolyte/blocking-electrode.⁴⁻⁷ Based on the EIS analysis of Au/solid electrolyte/Au, the conductivity of Au/Cathode without KB/Au was analyzed. The Z' value of the inflection point was determined as the total resistance R_t of the tested samples,^{6,7} and the conductivity of the cathodes were then calculated using $\sigma = \frac{1}{R} \frac{L}{A}$, where σ is the conductivity of the cathode; R is the total resistance R_t ; L and A are the cathode thickness and measurement area, respectively. The calculated results of the conductivity are shown in Fig. 1b.

Fig. 2 EIS analysis of cathodes without KB in the frequency range of 7 MHz~0.1 Hz. **a** EISs and **b** corresponding equivalent circuit of Au/ Cathode without KB/Au samples.

Additionally, according to the reviewer's suggestion, EIS measurement was performed at higher frequencies (7 MHz~0.1Hz). As shown in Fig. 2a, the impedance plot of Au/Cathode without KB/Au includes a semicircle at high frequency region and a tail at low frequency region. It is worth noting that compared with 1 MHz, the semicircle at 7 MHz shows more information about grain boundary resistance of LLZTO electrolyte, but the overall trend of the curves is not changed. The corresponding equivalent circuit is shown in Fig. 2b.⁷⁻⁹ R_b and R_g are for bulk and grain boundary resistance of KB-free cathodes plate, respectively. CPE_g is for double-layer capacitance at grain boundary. R_{el} and CPE_{el} are for the resistance of Au electrode and the interface between LLZTO and Au electrode. The total resistance R_t , almost equals to the Z' value of the inflection point mentioned above, is the sum of R_g and R_b , and was used to calculate the conductivity. The calculated results (Table 1) are consistent with the results tested in the frequency range of 1 MHz~0.1 Hz, showing that the LCO-LLZTO@LCO cathode exhibits an increased conductivity. Note that the electronic conduction is also not separated from ionic conduction here.

Table 1 Analysis results of cathodes without KB.

Cathodes without KB	R_b (Ω)	R_g (Ω)	CPE_g (10^{-9} F)	R_t (Ω)	Conductivity (10^{-5} S cm^{-1})
LCO-LLZTO@LCO	20	42	1.6	62	5.13
LCO-LLZTO@Li ₂ CO ₃	24	247	2.8	271	1.17
LCO	25	542	1.9	567	0.47

The following description has been made in new supporting information:

Fig. S10: Fig. S10a shows the impedance plots of Au/Cathode without KB/Au samples. The Z' value of the inflection point was determined as the total resistance R_t of the tested samples

Fig. S10: All the term "ionic conductivity" in the EIS analysis was corrected to "conductivity".

Comment 2: *The authors are to be commended for the TEM analysis as it is complex and laborious. They have satisfied the question regarding the quality and nature of the bond between LCO and LLZTO. Well done. However, the other part of this response was practical and more important in terms of the impact of this study. The quality of the interface regarding the resistance remains unanswered. If the authors haven't read it already, the work by Goodenough on LCO-LLZO indicated that Co diffuses into LLZO thereby compromising the interface kinetics. Thus, while the authors did a great analysis with the TEM, the question about the interface kinetics remains. Can the authors use micro-electrodes along with their nicely FIB'ed samples to achieve this?*

Response: We thank the reviewer for raising useful comments. We have now read and cited the article by Goodenough. The diffusion of Co element between LLZO and LCO above 700 °C, as reported, was also found in our experiments, but in our case there is no obvious element diffusion between Ta-doped LLZO and LCO at 600 °C. The stability of Ta-doped LLZO and LCO at 600 °C has also been reported previously,¹⁰ which is also one of the important factors for us to determine the sintering temperature. But we are very sorry for not being able to achieve the microelectrode testing due to the limitation of existing equipment and our experience. In view of this, electrochemical tests of the cold-pressed LLZTO, LLZTO@Li₂CO₃ and LLZTO@LCO pieces were undertaken to analyze the ion-transport properties at the LLZTO/LCO interface. This method minimizes the type of interface inside test samples by excluding the use of other additives, leading to the LLZTO/LCO interface that can be distinguished in EIS. As shown in Fig. 3, LLZTO, LLZTO@Li₂CO₃ and LLZTO@LCO powders were filled into the insulating bush and pressed into pieces at room temperature, respectively. Gold foils were used as the blocking electrodes. The EIS tests of the cold-pressed pieces were performed under 350 MPa and 80 °C.

Fig. 3 Physical photo and structure schematic of the cold-pressing mold. (This figure is used as Fig. S13a in supplementary information)

Fig. 4 Characterization of the cold-pressed LLZTO, LLZTO@Li₂CO₃ and LLZTO@LCO pieces. **a** EISs of Au/cold-pressed piece/Au samples. **b** Equivalent circuit of Au/LLZTO/Au. **c** Equivalent circuit of Au/LLZTO@LCO/Au and Au/LLZTO@Li₂CO₃/Au. (This figure is used as Fig. S13 in supplementary information)

Fig. 4a shows the normalized (thickness is taken into account) EISs of LLZTO, LLZTO@Li₂CO₃ and LLZTO@LCO cold-pressed pieces, respectively. The impedance plot of Au/LLZTO/Au contains an obvious semicircle at high frequency region and a tail at low frequency region. The corresponding equivalent circuit is shown in Fig. 4b.^{7, 8} R_b and R_g are for bulk and grain boundary resistance of cold-pressed piece, respectively. CPE_g is for double-layer capacitance at grain boundary. R_{el} and CPE_{el} are for the resistance of Au electrode and the interface between LLZTO and Au electrode. The fitting results are shown in the Table 2. It can be seen that LLZTO cold-pressed piece shows a grain boundary resistance of R_g (25057 Ω), which can be attributed to the LLZTO/LLZTO interface. However, the impedance plots of Au/LLZTO@LCO/Au and

Au/LLZTO@Li₂CO₃/Au show a flat semicircle at high frequency region, which is due to the presence of the LCO and Li₂CO₃ surface layer resulting in an additional poorly-resolved semicircle. This is consistent with the previously reported EIS changes of the lithium silicate glass after the formation of a hydrated/carbonated surface layer.¹¹ Their equivalent circuit is shown in Fig. 4c. R₁ and CPE₁ are for LLZTO/LCO interface in LLZTO@LCO particle or LLZTO/Li₂CO₃ interface in LLZTO@Li₂CO₃ particle. R₂ and CPE₂ are for LCO/LCO interface between LLZTO@LCO particles or Li₂CO₃/Li₂CO₃ interface between LLZTO@Li₂CO₃ particles. The fitting results are shown in Table 3. It can be seen that the LLZTO@Li₂CO₃ cold-pressed piece shows an interfacial resistance of R₁ (28500 Ω) and an interfacial resistance of R₂ (50252 Ω), corresponding to LLZTO/Li₂CO₃ and Li₂CO₃/Li₂CO₃ interfaces respectively, indicating slow ion-transport between LLZTO/Li₂CO₃ particles. In contrast, the LLZTO@LCO cold-pressed piece shows a smaller LLZTO/LCO interface resistance R₁ of 15524 Ω and LCO/LCO interface resistance R₂ of 29083 Ω, confirming that the converted LLZTO/LCO interface enables faster lithium-ion transfer.

Table 2 Fitting results of Au/ cold-pressed piece/Au.

Cold-pressed pieces	R _b (Ω)	R _g (Ω)	CPE _g (10 ⁻⁹ F)
LLZTO	19	25057	5.8

Table 3 Fitting results of Au/cold-pressed LLZTO@Li₂CO₃/Au.

Cold-pressed pieces	R _b (Ω)	R ₁ (Ω)	R ₂ (Ω)	CPE ₁ (10 ⁻⁹ F)	CPE ₂ (10 ⁻⁷ F)
LLZTO@LCO	20	16524	29083	3.3	0.23
LLZTO@Li ₂ CO ₃	27	28500	50252	9.6	5.3

The EIS analysis of the cold-pressed pieces demonstrates the accelerated ion transfer at the LLZTO/LCO interface. Herein, we beg the reviewer to forgive us for failing to supplement the microelectrode measurement. Using FIB sample as a microelectrode provides an accurate and feasible way to study the ion-transport kinetics between the surface layer and LLZO, which will be considered as our next research.

The following revision has been made in our manuscript:

Page 10, line 13: but also have a tight and low-impedance interface at the junction of LLZTO core and LiCoO₂ shell (Fig. S12, 13), promoting the transport of lithium ions inside the cathode.

References: Park, K. *et al.* Electrochemical nature of the cathode interface for a solid-state lithium-ion battery: Interface between LiCoO₂ and garnet-Li₇La₃Zr₂O₁₂. *Chem. Mater.* **28**, 8051-8059 (2016).

The following description has been made in new supporting information:

Fig. S13: Electrochemical tests of the cold-pressed LLZTO, LLZTO@Li₂CO₃ and LLZTO@LCO pieces were undertaken to analyze the ion-transport properties at the LLZTO/LCO interface. This method minimizes the type of interface inside test samples by excluding the use of other additives, lead to the LLZTO/LCO interface can be distinguished in EIS. As shown in Fig. 3, LLZTO, LLZTO@Li₂CO₃ and LLZTO@LCO powders were filled into the insulating bush and pressed into pieces at room temperature, respectively. Gold foils were used as the blocking electrodes. The EIS tests of the cold-pressed pieces were performed under 350 MPa and 80 °C. Fig. S13a shows the normalized (thickness is taken into account) EISs of LLZTO, LLZTO@Li₂CO₃ and LLZTO@LCO cold-pressed pieces, respectively. The impedance plot of Au/LLZTO/Au contains an obvious semicircle at high frequency region and a tail at low frequency region. The corresponding equivalent circuit is shown in Fig. 4b. R_b and R_g are for bulk and grain boundary resistance of cold-pressed electrolyte pieces, respectively. CPE_g is for double-layer capacitance at grain boundary. R_{el} and CPE_{el} are for the resistance of Au electrode and the double-layer capacitance at the interface between LLZTO and Au electrode. The fitting results are shown in Table 2. It can be seen that LLZTO cold-pressed piece shows a grain boundary resistance of R_g (25057 Ω), which can be attributed to the grain boundary of LLZTO/LLZTO. However, the impedance plots of Au/LLZTO@LCO/Au and Au/LLZTO@Li₂CO₃/Au show a flat semicircle at high frequency region, which is due to the presence of the LCO and Li₂CO₃ surface layer resulting in the addition of a poorly-resolved semicircle. Their equivalent circuit is shown in Fig. 4c. R₁ and CPE₁ are for LLZTO/LCO interface in

LLZTO@LCO particle or LLZTO/Li₂CO₃ interface in LLZTO@Li₂CO₃ particle. R₂ and CPE₂ are for LCO/LCO interface between LLZTO@LCO particles or Li₂CO₃/Li₂CO₃ interface between LLZTO@Li₂CO₃ particles. The fitting results are shown in Table 3. It can be seen that the LLZTO@Li₂CO₃ cold-pressed piece shows an interfacial resistance of R₁ (28500 Ω) and an interfacial resistance of R₂ (50252 Ω), corresponding to LLZTO/Li₂CO₃ and Li₂CO₃/Li₂CO₃ interfaces respectively, indicating slow ion-transport between LLZTO/Li₂CO₃ particles. In contrast, the LLZTO@LCO cold-pressed piece shows a smaller LLZTO/LCO interface resistance R₁=15524 Ω and LCO/LCO interface resistance R₂=29083 Ω, confirming that the converted LLZTO/LCO interface enables faster lithium-ion transfer.

Table 1 Fitting results of Au/ cold-pressed piece/Au.

Cold-pressed pieces	R _b (Ω)	R _g (Ω)	CPE _g (10 ⁻⁹ F)
LLZTO	19	25057	5.8

Table 2 Fitting results of Au/cold-pressed LLZTO@Li₂CO₃/Au.

Cold-pressed piece	R _b (Ω)	R ₁ (Ω)	R ₂ (Ω)	CPE ₁ (10 ⁻⁹ F)	CPE ₂ (10 ⁻⁷ F)
LLZTO@LCO	20	16524	29083	3.3	0.23
LLZTO@Li ₂ CO ₃	27	28500	50252	9.6	5.3

Comment 3: 1) *Is it correct to assume Li metal was one electrode? If so, where does the Li interface resistance show up in the EIS analysis.*

Response: As mentioned by the reviewer, Li/SN/LLZTO/SN/LCO-LLZTO@LCO cells with a lithium metal anode were used for EIS analysis. But, the Li anode interface resistance and the cathode interface resistance were not distinguished in EIS, which is also present in the Li/gel/SSE/gel/LiFePO₄ and Li/SN-FEC/LLZTO/SN-FEC/LiFePO₄ cells in previous reports.^{12, 13} As they stated in the article, the resistance of a semicircle in the middle frequency region is the sum of Li anode interface resistance and cathode interface resistance, so a resistor R_{int} and a constant phase element CPE_{int} were used to equivalent the total interface resistance of anode and cathode.

2) *Why do the grain boundary and interface resistances change for both cells?*

3) *The @lithium carbonate cell EIS data seems to change in the EIS data between the bulk and grain boundary. Why is this so?*

Response: As the reviewer stated, the fitting results show that the grain boundary and bulk resistance of the cells were increased slightly. The increase in grain boundary and bulk resistance may be related to the micro-cracks generated in the LLZTO electrolyte. As shown in Fig. 5a, the LLZTO piece newly prepared by the hot-pressing method exhibits a uniform surface and high density (>99%). But after long-term cycling, obvious cracks sometimes appear on the LLZTO electrolyte piece (Fig. 5b), which may be the cause of the increase in grain boundary and bulk resistance, but the significant impact on the normal operation of the cells was not found. The reason for cracking is not yet clear and further research is needed.

Fig. 5 Cracks in LLZTO electrolyte piece after long-term cycling. **a** Optical photo of the new LLZTO piece. **b** Optical photo of the LLZTO piece after long-term cycling.

The increase in total interface resistance, on the one hand, comes from the

increase in the electrode/LLZTO interface resistance due to the instability of the SN buffer layer, and the increase in the interface resistance between the particles inside cathode due to the volume change during long-term cycling, on the other hand. In the @Li₂CO₃ cathode, the decomposition of Li₂CO₃ aggravates the cracking of the interface between the particles inside cathode, thus exhibiting a larger cathode interface resistance.

4) Can the authors justify why they used this equivalent circuit or who they will cite for this equivalent circuit? The bulk resistance requires a capacitive or constant-phase-element otherwise there would be no capacitive signal or imaginary impedance values.

Response: As shown in Fig. 6, the impedance plot includes an incomplete semicircle in the high frequency region (~1M Hz), a semicircle in the middle frequency region (~794 Hz) and a tail in low frequency region (~0.1 Hz). The corresponding equivalent circuit is shown in the inset of Fig. 6. R_b and R_g are for bulk and grain boundary resistance of LLZTO electrolyte piece, respectively. R_{int} and CPE_{int} are for total interface resistance and double-layer capacitance. The Warburg impedance (W) and CPE_d are for diffusion impedance of electrode.

Fig. 6 EIS analysis and corresponding equivalent circuit of the cell in our article.

In the high frequency region (>0.1 MHz), ideally there will be two semicircles corresponding to the bulk and grain boundary resistance, respectively.¹⁴ But in the measurement, depending on the sample, the semicircle corresponding to the bulk resistance often does not appear.^{9, 15} In this case, the R_b(R_gCPE_g) circuit model is

generally used to equivalent the bulk and grain boundary resistance of the electrolyte.^{7, 8, 12, 16} In addition, it is also reported that the high-frequency intersection of the semicircles with the real axis is the bulk resistance of LLZTO electrolyte and the bulk resistance was equivalent to a resistor R_b .¹⁷ Therefore, considering that only one incomplete semicircle appears in the high frequency region in our EIS, a resistor R_b connected in series with paralleled R_g and CPE_g was used to equivalent the bulk resistance and grain boundary resistance of the LLZTO electrolyte.

The semicircle in the middle frequency region (0.1 MHz~10Hz) corresponds to the total interface resistance, including anode interface resistance and cathode interface resistance.^{12, 13, 18} These interfaces could not be distinguished, so they were equivalent to a resistor R_{int} and a constant phase element CPE_{int} . In low frequency region (10 Hz~0.1 Hz), The tail deviated by 45° was equivalent to a Warburg impedance (W) and a constant phase element CPE_d ,¹² corresponding to the diffusion impedance of electrode.

Therefore, the $R_b(R_gCPE_g)(R_{int}CPE_{int})(CPE_dW)$ circuit model was used to analyze the interface resistance of the cells in our experiment.

References:

- 1 Saito, T., & Uosaki, K. Surface film formation and lithium underpotential deposition on Au(111) surfaces in propylene carbonate: In situ scanning tunneling microscopy study. *J. Electrochem. Soc.* **150**, A532- A537 (2003).
- 2 Murugan, R., Thangadurai, V. & Weppner, W. Fast lithium ion conduction in garnet-type $Li_{(7)}La_{(3)}Zr_{(2)}O_{(12)}$. *Angew. Chem. Int. Ed. Engl.* **46**, 7778-7781 (2007).
- 3 Thangadurai, V., Robert A. Huggins, and W. Weppner. Use of simple ac technique to determine the ionic and electronic conductivities in pure and Fe-substituted $SrSnO_3$ perovskites. *J. power sources* **108**, 64-69 (2002).
- 4 Liu, W. *et al.* Ionic conductivity enhancement of polymer electrolytes with ceramic nanowire fillers. *Nano. Lett.* **15**, 2740-2745 (2015).
- 5 Zhang, X. *et al.* Synergistic coupling between $Li_{6.75}La_3Zr_{1.75}Ta_{0.25}O_{12}$ and

- poly(vinylidene fluoride) induces high ionic conductivity, mechanical strength, and thermal stability of solid composite electrolytes. *J. Am. Chem. Soc.* **139**, 13779-13785 (2017).
- 6 Huang, X. *et al.* Influence of $\text{La}_2\text{Zr}_2\text{O}_7$ additive on densification and Li^+ conductivity for Ta-doped $\text{Li}_7\text{La}_3\text{Zr}_2\text{O}_{12}$ garnet. *Jom.* **68**, 2593-2600 (2016).
 - 7 Li, Y., Han, J., Wang, C., Xie, H. & Goodenough, J.B. Optimizing Li^+ conductivity in a garnet framework. *J. Mater. Chem.* **22**, 15357 (2012).
 - 8 Huang, X., Xiu, T., Badding, M.E. & Wen, Z. Two-step sintering strategy to prepare dense Li-Garnet electrolyte ceramics with high Li^+ conductivity. *Ceram. Int.* **44**, 5660-5667 (2018).
 - 9 Xiang, X., Chen, F., Shen, Q., Zhang, L. & Chen, C. Effect of the lithium ion concentration on the lithium ion conductivity of Ga-doped LLZO. *Mater. Res. Express* **6**, 085546 (2019).
 - 10 Ren, Y., Liu, T., Shen, Y., Lin, Y. & Nan, C. W. Chemical compatibility between garnet-like solid state electrolyte $\text{Li}_{6.75}\text{La}_3\text{Zr}_{1.75}\text{Ta}_{0.25}\text{O}_{12}$ and major commercial lithium battery cathode materials. *J. Materiomics* **2**, 256-264 (2016).
 - 11 Irvine, J., Sinclair, D. & West, A. Electroceramics: characterization by impedance spectroscopy. *Adv. Mater.* **2**, 132-138 (1990).
 - 12 Liu, B. *et al.* Garnet solid electrolyte protected Li-metal batteries. *ACS Appl. Mater. Interfaces* **9**, 18809-18815 (2017).
 - 13 Lu, Z. *et al.* Enabling room-temperature solid-state lithium-metal batteries with fluoroethylene carbonate-modified plastic crystal interlayers. *Energy Storage Mater.* **18**, 311-319 (2018).
 - 14 Sharafi, A. *et al.* Impact of air exposure and surface chemistry on $\text{Li}-\text{Li}_7\text{La}_3\text{Zr}_2\text{O}_{12}$ interfacial resistance. *J. Mater. Chem. A* **5**, 13475-13487 (2017).
 - 15 Pfenninger, R., Struzik, M., Garbayo, I., Stilp, E. & Rupp, J.L.M. A low ride on processing temperature for fast lithium conduction in garnet solid-state battery films. *Nat. Energy* **4**, 475-483 (2019).
 - 16 Lu, Y., Meng, X., Alonso, J.A., Fernandez-Diaz, M.T. & Sun, C. Effects of fluorine doping on structural and electrochemical properties of $\text{Li}_{6.25}\text{Ga}_{0.25}\text{La}_3\text{Zr}_2\text{O}_{12}$ as

electrolytes for solid-state lithium batteries. *ACS Appl. Mater. Interfaces* **11**, 2042-2049 (2018).

- 17 Deng, T. *et al.* Tuning the anode-electrolyte interface chemistry for garnet-based solid-state Li metal batteries. *Adv. Mater.* **32**, 2000030 (2020).
- 18 Subramani, R. *et al.* High Li⁺ transference gel interface between solid-oxide electrolyte and cathode for quasi-solid lithium-ion batteries. *J. Mater. Chem. A* **7**, 12244-12252 (2019).